# Downregulation of praja2 restrains endocytosis and boosts tyrosine kinase receptors in kidney cancer
Laura Rinaldi[1], Francesco Chiuso[1], Emanuela Senatore[1], Domenica Borzacchiello[1], Luca Lignitto[2], Rosa Iannucci[1], Rossella Delle Donne[1], Mariano Fuggi[3], Carla Reale [4], Filomena Russo[4], Nicola Antonino Russo [4], Giorgio Giurato [5,6], Francesca Rizzo [5,6], Assunta Sellitto[6], Michele Santangelo[3], Davide De Biase[7], Orlando Paciello[8], Chiara D'Ambrosio [9], Stefano Amente[1], Corrado Garbi[1], Emiliano Dalla [10], Andrea Scaloni[9], Alessandro Weisz[5,6], Concetta Ambrosino [4,11], Luigi Insabato[3] & Antonio Feliciello [1] ✉

Clear cell renal cell carcinoma (ccRCC) is the most common kidney cancer in the adult population. Late diagnosis, resistance to therapeutics and recurrence of metastatic lesions account for the highest mortality rate among kidney cancer patients. Identifying novel biomarkers for early cancer detection and elucidating the mechanisms underlying ccRCC will provide clues to treat this aggressive malignant tumor. Here, we report that the ubiquitin ligase praja2 forms a complex with-and ubiquitylates the AP2 adapter complex, contributing to receptor endocytosis and clearance. In human RCC tissues and cells, downregulation of praja2 by oncogenic miRNAs (oncomiRs) and the proteasome markedly impairs endocytosis and clearance of the epidermal growth factor receptor (EGFR), and amplifies downstream mitogenic and proliferative signaling. Restoring praja2 levels in RCC cells downregulates EGFR, rewires cancer cell metabolism and ultimately inhibits tumor cell growth and metastasis. Accordingly, genetic ablation of praja2 in mice upregulates RTKs (i.e. EGFR and VEGFR) and induces epithelial and vascular alterations in the kidney tissue.

In summary, our findings identify a regulatory loop between oncomiRs and the ubiquitin proteasome system that finely controls RTKs endocytosis and clearance, positively impacting mitogenic signaling and kidney cancer growth.

Renal carcinoma (RCC) is the most frequent renal neoplasia and, along with bladder carcinoma and prostate cancer, is one of the three most prevalent genitourinary system tumors[1,2]. In the last issue of WHO, 16 subtypes of renal tumors have been recognized[3]. Clear cell renal cell carcinoma (ccRCC) is a morphologically heterogeneous group of malignant neoplasms that represents about 75% of the kidney tumors, with 30% of patients presenting localized ccRCC that eventually develop metastasis and require systemic treatment. RCC is inherited in 3–5% of cases and there are various genetic syndromes that increase the risk of RCC development, including Von Hippel Lindau disease (VHL) and Tuberous sclerosis complex[4,5]. Molecular analysis of genetic disorders linked to the development of kidney neoplasia identified some of the pathogenic mechanisms underlying RCC, although in the majority of cases the mechanism(s) remains largely unknown. In particular, deregulation of VHL function, either by inactivating mutations or

[1]Department of Molecular Medicine and Medical Biotechnology, University Federico II, Naples, Italy. [2]Cancer Research Center of Marseille (CRCM), CNRS, Aix Marseille Univ, INSERM, Institut Paoli-Calmettes, Marseille, France. [3]Department of Advanced Biomedical Sciences, University Hospital Federico II, Naples, Italy. [4]Biogem, Biology and Molecular Genetics Institute, Ariano Irpino, Italy. [5]Genome Research Center for Health, Baronissi (SA), Italy. [6]Laboratory of Molecular Medicine and Genomics, Department of Medicine, Surgery and Dentistry SMS, Baronissi (SA), Italy[7]Department of Pharmacy, University of Salerno, Salerno, Italy. [8]Present address: Department of Veterinary Medicine and Animal Production, Pathology Unit, University Federico II, Naples, Italy. [9]Proteomics, Metabolomics and Mass Spectrometry Laboratory, ISPAAM, National Research Council, Portici (Naples), Italy. [10]Department of Medicine, University of Udine, Udine, Italy. [11]Department of Science and Technology University of Sannio, Sannio, Italy. ✉e-mail: feliciel@unina.it

promoter hypermethylation, is a common event in ccRCC, whereas germline mutations of this gene in the VHL syndrome cause hundreds of pre-neoplastic renal cysts in affected patients[6]. In both cases, the inactivation of VHL gene function leads to the accumulation of the hypoxia-induced factor α (HIF1α) and consequent upregulation of hypoxia-responsive genes, including vascular endothelial growth factor A (VEGF-A)[7]. Accordingly, ccRCC lesions show high expression levels of vascular VEGF-A and its receptors, including VEGFR2, which currently represent actionable therapeutic targets[8].

The expression of tyrosine kinase receptors (RTKs) has been found altered in different types of RCC[9,10]. In particular, the levels of c-Met oncogene are abnormally elevated in ccRCC tissues and are linked to a poor survival rate in patient[11]. c-Met overexpression induces downstream activation of mitogenic signaling cascades and PI3K•Akt axis, which lead to cancer cell survival and uncontrolled proliferation. Mutations within the c-Met kinase domain identified as causative of renal papillary carcinoma trigger constitutive receptor activation. Mechanistically, c-Met overactivation is induced by its abnormal accumulation in endosomes from which it continues to signal to downstream oncogenic pathways[12]. Blockage of endocytosis reduces the endosomal localization of the receptor, thus inhibiting cell growth and anchorage-independent proliferation. Similarly, EGFR upregulation has been described in chronic inflammatory kidney diseases characterized by proliferative disorders, such as polycystic kidney disease[13]. Proof-of-concept studies demonstrated that inhibition of EGFR can ameliorate the renal fibrotic damage, exerting beneficial effects on the otherwise progressive kidney disease. These and other studies established a key role for RTKs expression and activation in signal transduction events coupled to nuclear gene expression, metabolic rewiring, and kidney cancer cell growth and proliferation[14].

Clathrin-mediated endocytosis is the key mechanism that regulates recycling and degradation of activated membrane receptors. Dysregulation of proteins involved in the endocytic pathway leads to excessive activation of mitogenic pathways and cell transformation[15–17]. The endocytic fate of membrane receptors, including EGFR, is tightly controlled by the ubiquitin pathway[18–20]. Upon ligand stimulation, EGFR is ubiquitinated by the E3 ubiquitin ligase Cbl. EGFR ubiquitination allows the recruitment of a molecular complex constituted by Epidermal growth factor receptor substrate 15 (Eps15), Epsin, Ral-Binding Protein 1 (RalBP1), BTB/POZ domain-containing protein 1 (POB1), and adapter protein 2 (AP2). The formation of this complex targets the receptor to the endocytic route, eventually leading to its lysosome-assisted degradation[21]. Deubiquitination of endocytosed receptors by ubiquitin-specific proteases (USPs) eventually results in recycling the receptor back to the membrane[22–25].

Adapter protein complexes (APs), AP1 and AP2, organize the assembly of clathrin-coated vesicles, allowing the binding between clathrin moieties, vesicle membranes, and cargos[26]. AP2 is the major hub for the maturation of endocytic vesicles and consists of 4 subunits: α and β2 adaptins, μ2 and σ2. In response to endocytic stimuli, such as RTK ligands, AP2 binds to phosphatidylinositol 4,5-bisphosphate and undergoes conformational changes that enable the interaction of the hydrophobic pocket of μ2 subunit with the cytoplasmic tail of transmembrane protein cargos and the clathrin coat polymerization[27,28]. Conformational rearrangements of AP2 in course of RTK endocytosis are tightly regulated by reversible phosphorylation[29]. However, phosphorylation of the μ2 subunit does not explain the constitutive (ligand-independent) endocytosis, suggesting that other post-translational mechanisms, including ubiquitylation, are required to activate AP2[30].

Praja2 is an E3 ligase widely expressed in mammalian cells and tissues which is involved in essential aspects of cell signaling. Praja2 efficiently links phosphorylation to ubiquitination of protein kinases, scaffolds, and effectors, with important implications for development, inflammatory responses, neuronal activity, primary ciliogenesis, cancer cell growth, and metabolism[31–39]. Dysregulation of praja2-regulated pathways has been causally linked to the growth and progression of human glioblastoma and gastric cancer[40,41]. However, the impact of praja2 on clathrin-mediated receptor endocytosis and downstream signaling, and its impact in human cancer were unexplored.

Here, we found that praja2 is a component of the clathrin-coated vesicles and an essential regulator of receptor endocytosis and signaling. A regulatory loop between oncomiRs and the proteasome machinery operating in ccRCC cells controls praja2 levels, markedly impacting on RTKs endocytosis and tumor growth.

## Results

### Praja2 binds to- and ubiquitinates AP2m

To gain insight into praja2 function, we performed a yeast two-hybrid screening using a human cDNA library and the C-terminal region of praja2 as bait. This screening allowed the identification of a C-terminal portion of the adapter-related protein complex 2 subunit μ2 (AP2m) as a putative praja2-interacting protein (Supplementary Fig. 1a). AP2m is a component of the heterotetrameric coat assembly protein complex 2 (AP2) that is required for receptor endocytosis and signaling[27,28,42]. These data suggest that praja2 may be a novel component of the endocytic pathway and a potential regulator of the intracellular vesicular trafficking. To further address this issue, we performed a proteomic analysis of affinity-purified praja2 complexes from total cell lysates. Specifically, HEK293 cell lysates overexpressing FLAG-praja2 inactive mutant RM or FLAG-Empty Vector (EV) were subjected to immunoprecipitation.

Proteomic analysis of the precipitates identified multiple praja2 potential interactors. We then used our interactomic analysis of praja2 to build a protein-protein interaction (PPI) network. Specifically, we queried the inBio Discover web tool to create an experimental-based network that included 506 proteins (out of the 724 that constituted the input list) which were known to physically interact with each other. Next, we performed a Gene Ontology: Biological Process functional enrichment analysis of this PPI network, focusing on the subnetwork associated with endocytosis and membrane trafficking, recapitulated by the following functional terms: endocytosis, vesicle-mediated transport, regulation of autophagy, and Golgi vesicle transport. Our analysis revealed a remarkable number of proteins ($n = 87$) that were associated with at least one of these terms, demonstrating that the praja2 interactome is deeply involved in vesicular trafficking (Fig. 1a, Supplementary Data 1). We decided to study in detail the interaction of the AP2m protein with praja2 because it regulates receptor trafficking and endocytosis, in which praja2 may be involved. Co-immunoprecipitation assays confirmed that FLAG tagged-praja2 and HA tagged-AP2m form a stable complex in cells (Fig. 1b). Deletion mutagenesis combined with co-immunoprecipitation experiments demonstrated that AP2m interacts with the segment spanning the aminoacids 430–530 in praja2 (Fig. 1b). EGF induces a rapid recruitment of the AP2 complex in close proximity of the plasma membrane, favoring endocytic vesicles formation and receptor internalization[43]. Accordingly, we evaluated the intracellular localization of praja2 and AP2m by in situ immunostaining analysis both in HK-2 and HeLa cells stimulated with EGF (10 ng/ml). The cells were placed on ice for 30 min and then incubated at 37 °C to induce the formation of clathrin-coated vesicles. We found that endogenous praja2 and AP2m colocalized in the endocytic vesicles in EGF-stimulated cells (Fig. 1c, d, Supplementary Fig. 1b, c), indicating that praja2 and AP2m form a complex in intact cells. We also evaluated the temporal dynamics of this complex in cells stimulated with EGF. Figure 1e, f show that EGF induced a rapid and transient formation of the praja2/AP2m complex. Since praja2 is an E3 ubiquitin ligase that ubiquitinates a variety of cellular substrates[37,44], we tested whether praja2 ubiquitinated AP2m in vivo. Overexpression of wild-type praja2, but not its inactive mutant praja2RM, induced a robust accumulation of poly-ubiquitinated AP2m (Fig. 1g). Moreover, EGF induced a poly-ubiquitination of AP2m which was prevented by silencing of praja2 (Fig. 1h). We also monitored ubiquitination of AP2m using antibodies directed against distinct Lysines residues (K48 e 63) of Ubiquitin. The results show only a mild enhancement of AP2m ubiquitination by co-expressed praja2 at both sites (Supplementary Fig. 2a, b).

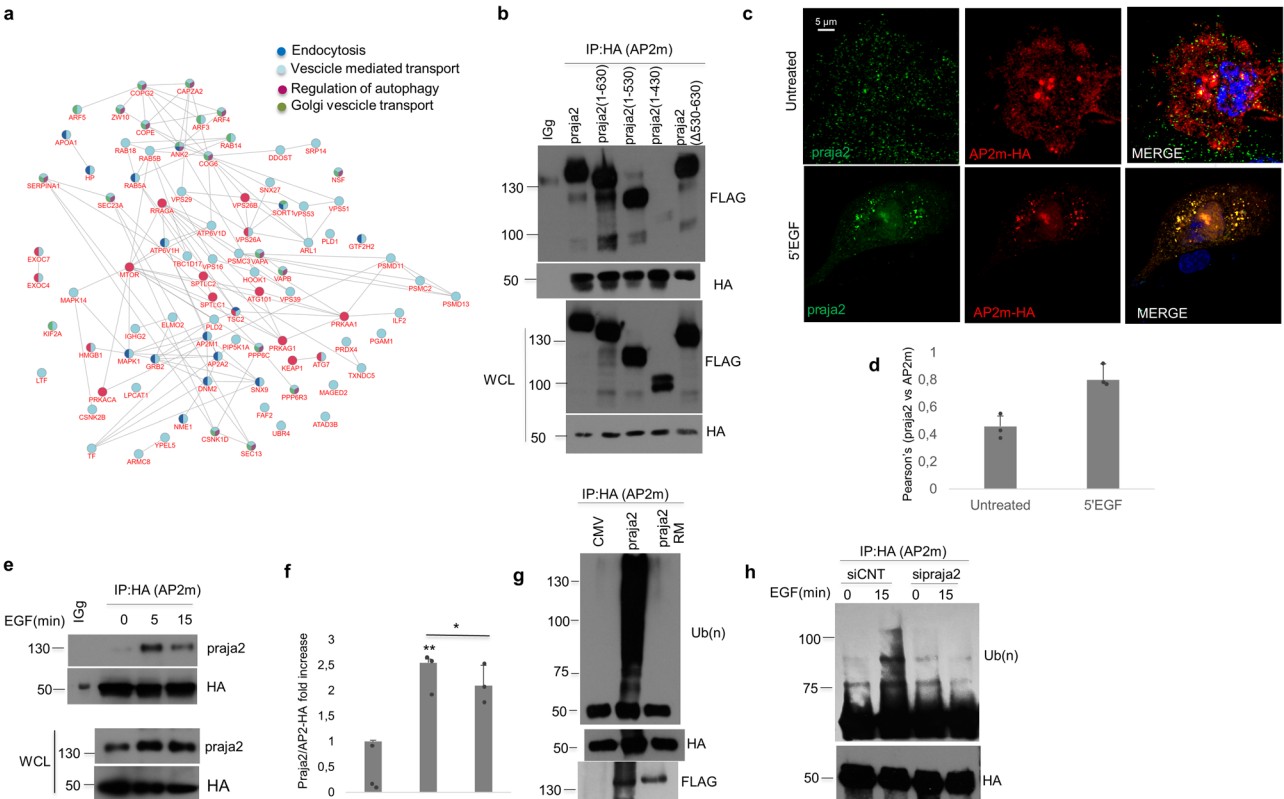

**Fig. 1 | Praja2 interacts with- and ubiquitinates AP2m subunit. a** Subnetwork of the praja2 protein-protein interactome (PPI) associated with autophagy and endocytosis. After experimentally defining the full praja2 PPI network ($n = 724$) using the inBio Discover Web tool, a functional enrichment analysis was performed focusing specifically on terms associated with autophagy and endocytosis, thus identifying a subgroup of proteins ($n = 87$) whose activities also included more specific functions related to Golgi vesicle transport. **b** Co-immunoprecipitation of AP2m-HA and FLAG-praja2 from HEK293 cell lysates expressing praja2 wild-type and deletion mutants (1-630; 1-530; 1-430; Δ530-630). The immunoprecipitation (Ip) was performed using anti-HA antibody or control IgG. **c** HK-2 cells, left untreated or treated with EGF, were PFA-fixed and immunostained for praja2 and AP2m. Representative confocal images are shown. Scale bar (5 µM). **d** Pearson's coefficient and quantitative analysis of the experiments shown in (**c**). A mean value of three independent experiments ± SD is shown. *$P = 0.014$. **e** HeLa cells were

transiently transfected with AP2m-HA, starved overnight, treated with EGF (10 ng/µl) for the indicated time points, and lysed. Lysates were immunoprecipitated with anti-HA antibody. Precipitates and an aliquot of lysates were immunoblotted with anti-praja2 and anti-HA antibodies. **f** Quantitative analysis of the experiments shown in (**e**). A mean value of three independent experiments ± SD is shown. Student's $t$-test, **$P = 0.004$; *$P = 0.01$. **g** Immunoprecipitation of AP2m-HA from HEK293 cell lysates expressing AP2m-HA, Myc-ubiquitin, and FLAG-praja2 or FLAG-praja2RM (inactive mutant). Precipitates were immunoblotted with anti-Myc (ubiquitinated AP2m) and anti-HA antibodies. FLAG-praja2 expression was revealed from total lysates. **h** Lysates from HeLa cells transiently transfected with Myc-ubiquitin, AP2m-HA, and siRNAs (control or targeting praja2), left untreated or stimulated with EGF (10 ng/µl) for 15 min, were subjected to immunoprecipitation with anti-HA antibody. Precipitates were immunoblotted with anti-HA and anti-Myc antibodies.

## Praja2 is required for receptor endocytosis

The data above suggest that praja2 is a component of the AP2-mediated protein complex assembled at the plasma membrane and a potential regulator of receptor endocytosis and signaling. To confirm this hypothesis, we monitored EGF receptor (EGFR) endocytosis by immunostaining analysis in serum-deprived HeLa cells left untreated or stimulated with EGF. As shown in Fig. 2a, b, in control cells, EGF treatment induced a rapid internalization of EGFR, as pointed out by a marked reduction of the signal at the cell membrane that was mostly re-localized within cytoplasmic endocytic vesicles. In contrast, in praja2-silenced HeLa cells, the EGFR remained localized at the plasma membrane and was not internalized as in the control cells. Immunoblot analysis of lysates from treated HeLa (Fig. 2c, d) and HK-2 (Supplementary Fig. 3a) cells shows that EGF induced a time-dependent reduction of EGFR levels. Similarly, praja2 levels were downregulated in EGF-treated cells (Fig. 2c). In contrast, depletion of praja2 increased the basal levels of EGFR and inhibited EGFR proteolysis in stimulated cells, with no major effects on AP2m levels (Fig. 2c).

AP2-mediated receptor endocytosis is also a common route for non-tyrosine kinase membrane receptors, such as the transferrin receptor (TfR). TfR is a receptor mediating the import of the transferrin-iron complex from the extracellular compartment to the cell by receptor-mediated

endocytosis[29,45]. After releasing the iron within the cell, TfR is rapidly transported back to the plasma membrane through the endocytic recycling system. To investigate if praja2 is also involved in transferrin endocytosis, we monitored the internalization of TfR in the presence of transferrin in serum-deprived HeLa cells. Figure 2e, f shows that, under basal conditions, TfR staining was mostly localized at the plasma membrane. Transferrin induced the internalization of TfR and the consequent accumulation of the receptor within the endocytic vesicles. In contrast, in praja2-silenced cells, TfR remained in the membrane in the presence of transferrin, indicating a severe impairment of TfR endocytosis. Immunoblot analysis confirmed that TfR levels were stable and the receptor on the membrane was not downregulated upon transferrin binding, both in control and in praja2-silenced cells (Supplementary Fig. 3b). The inhibition of transferrin-TfR endocytosis in the absence of praja2 was due to the loss of AP2m at the membrane. In fact, genetic knockdown of praja2 inhibited membrane localization of AP2m induced by EGF (Fig. 2g).

## A regulatory loop links EGFR and praja2

The data above suggest the existence of a regulatory loop that links RTK stimulation to praja2 turnover. We tested this hypothesis by measuring praja2 half-life in growing cells treated with the protein translation inhibitor

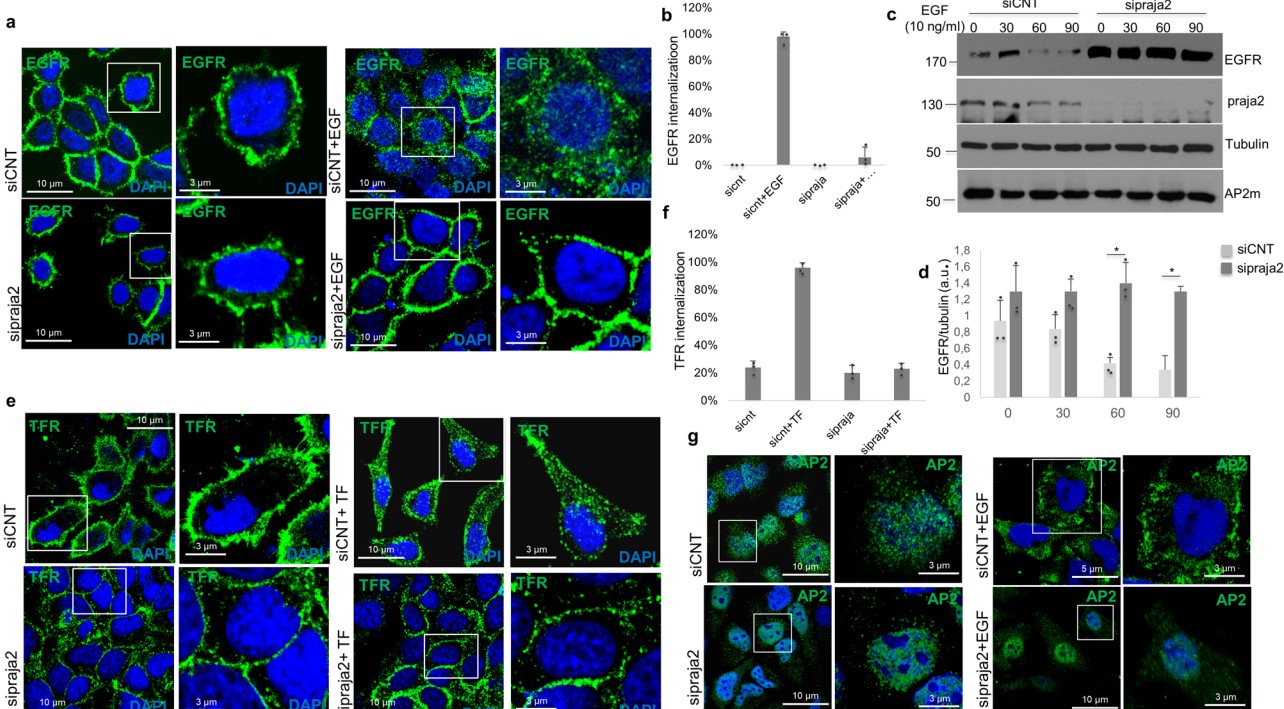

**Fig. 2 | Praja2 is required for receptor endocytosis. a** HeLa cells were transfected with siRNAs (control or targeting praja2), serum-deprived overnight, and treated with EGF (10 ng/µl) for 90 min. Cells were PFA-fixed and immunostained with anti-EGFR antibody (green) and DAPI (blue). Representative confocal images are shown. Scale bar is indicated. **b** Quantitative analysis of experiment shown in (**b**). A mean value of three independent experiments ± SD is shown. **c** Same as in (**a**), with the exception that cells were lysed and immunoblotted for EGFR, praja2, and α-tubulin. **d** Quantitative analysis of experiment shown in (**b**). A mean value of three independent experiments ± SD is shown. Student's *t*-test *P = 0.02 for 60 min, *P = 0.014 for 90 min time point. **e** HeLa cells were transfected with siRNAs (control

or targeting praja2), serum-deprived overnight, and treated with transferrin (Tf, 2 µg/ml) for 60 min. Cells were PFA-fixed and immunostained with anti-TfR antibody (green) and DAPI (blue). Representative confocal images are shown. Scale bar is indicated. **f** Quantitative analysis of experiment shown in (**b**). A mean value of three independent experiments ± SD is shown. **g** HeLa cells were transfected with siRNAs (control or targeting praja2), serum-deprived overnight, and treated with EGF (10 ng/µl) for 5 min. Cells were PFA-fixed and immunostained with anti-AP2m antibody (green) and DAPI (blue). Representative confocal images are shown. Scale bar is indicated.

cycloheximide. As shown in Fig. 3a, b, the half-life of wild-type praja2 protein is about 90 min. In contrast, the catalytic inactivation of praja2 (RING mutant, praja2RM) significantly stabilized the protein levels. The stabilization of the catalytically inactive praja2 mutant was not unexpected because activation of the E3 ligases frequently leads to their proteolysis[46]. Accordingly, photo-crosslinking activity-based protein profiling experiments show that endogenous praja2 activity was stimulated by physiological concentrations of EGF[47]. Next, we tested the effects of EGF stimulation on praja2 stability. As shown in Fig. 3c, d, EGF treatment induced a time-dependent decline of praja2 that was reversed by pre-treating the cells with the proteasome inhibitor MG132, confirming the existence of a feedback control mechanism based on praja2 proteolysis induced by EGF stimulation. In kidney cancer cells (A-498 and SN12C), the negative loop linking EGFR and praja2 appears to be amplified. In these cells, praja2 levels were extremely low, compared to non-tumoral epithelial kidney cells (HEK293), and inversely correlate with those of EGFR levels in both cell lines (Fig. 3e, f). Accordingly, transient overexpression of praja2 in both A-498 (Fig. 3g, h) and Caki-1 (Fig. 3i, j) kidney cancer cells markedly downregulated EGFR levels. The negative feedback between EGFR and praja2 finely controls the strength and duration (shape) of the transduction signals. Conversely, in RCC cells this autoregulatory loop is lost due to praja2 downregulation, thus leading to amplification of EGFR signals.

## Loss of praja2 in high-grade clear cell renal cell carcinoma
Clear cell renal cell carcinoma (ccRCC) represents about 80% of all kidney carcinomas[48,49]. EGFR overexpression correlates with invasiveness, malignant behavior and resistance to therapy of the majority of this type of cancer[50]. The data above indicate that downregulation of praja2 inhibits

EGF-induced receptor endocytosis and increases the levels of EGFR by several folds above control values. Based on this evidence, we analyzed the levels of praja2 in 53 ccRCC tumor tissues isolated from patients undergoing surgical nephrectomy. The data show that praja2 staining was reduced in 45 samples (Supplementary Data 2). In 27 out of 53 cases, the differential expression of praja2 between normal and tumor tissues was evaluated. As shown in Fig. 4a, praja2 staining was significantly and markedly reduced in tumor tissues, compared to normal counterparts (P = < 0.05; test di Mann–Whitney for praja2). We further addressed this issue by evaluating the expression of praja2 and EGFR in kidney cancer lesions and in the surrounding normal tissue. Figure 4b, c show that praja2 was significantly downregulated in high-grade lesions, while higher levels of the protein were detected in low-grade tumors and normal tissues. Conversely, EGFR levels were markedly upregulated in high-grade lesions compared to control tissues and to low-grade tumors. EGFR staining in high-grade lesions showed a prominent membrane localization, suggesting an impairment in the internalization of the receptor as observed in HeLa cells devoid of praja2. Immunoblot analysis performed on lysates from tumor samples of three different patients showed low levels of praja2 in tumor lesions compared to surrounding normal tissues (Fig. 4d, e). Conversely, in tumor samples the levels of EGFR and of the activating phosphorylation of p42/p44 mitogen-activated protein kinases at Thr202/Tyr204 (ERK1/2) were substantially higher compared to normal samples (Fig. 4d, e).

To understand the mechanism underlying praja2 downregulation in ccRCC samples, we monitored praja2 mRNA levels in Kidney Renal Clear Cell Carcinoma (KIRC, *n* = 533) and control (*n* = 72) samples using the UALCAN data-mining RNA-seq platform from TGCA repository[51,52]. The data show only a slight decrease of praja2 mRNA in KIRC tissue samples

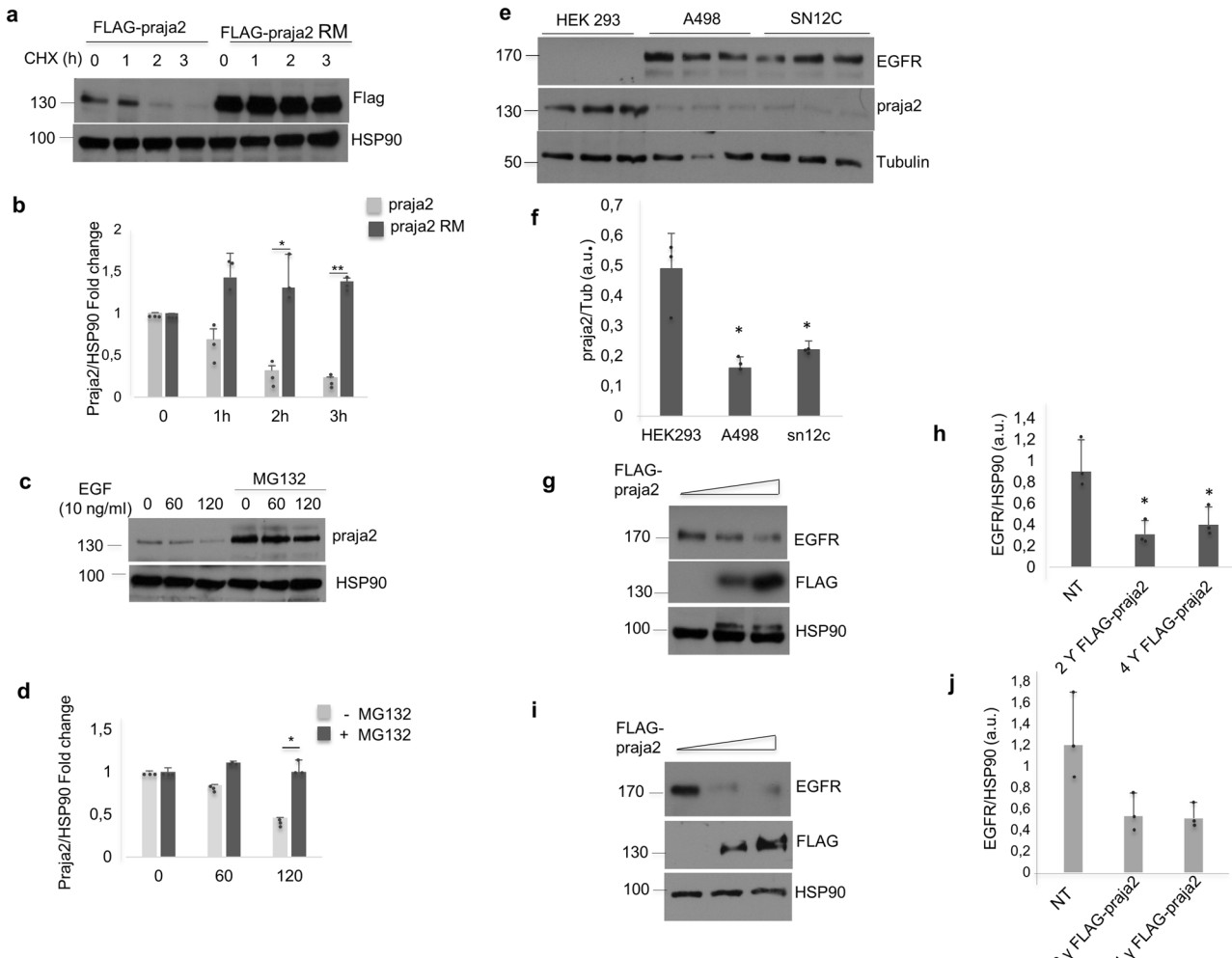

**Fig. 3 | EGF induces proteasomal degradation of praja2. a** HeLa cells transiently transfected with FLAG-praja2 (either wild type or ring mutant RM) vector were treated with cycloheximide (10 μM) and harvested at the indicated time points. Total lysates were immunoblotted with anti-FLAG. HSP90 was used as loading control. **b** Quantitative analysis of the experiments shown in a. A mean value of three independent experiments ± SD is presented. Student's *t*-test, *P = 0.02; **P = 0.001. **c** HeLa cells were serum-deprived overnight and stimulated with EGF (10 ng/ml) in the presence of the protein synthesis inhibitor cycloheximide (10 μM). Where indicated, cells were pre-treated with the proteasome inhibitor MG132 (10 μM). Cells were harvested at the indicated time points and lysed. Lysates were immunoblotted for praja2 and HSP90. **d** Quantitative analysis of the experiments shown in (**c**). A mean value of three independent experiments ± SD is shown. Student's *t*-test *P = 0.039. **e** Immunoblot analysis of praja2 and EGFR in HEK293 cells and in two

different kidney cancer human cell lines (A-498 and SN12C). Tubulin was used as loading control. **f** Quantitative analysis of the experiment shown in (**e**). A mean value of three independent experiments ± SD is shown. Student's *t*-test for A-498 *P = 0.02; for SN12C *P = 0.047 (**g**) A-498 cells were transiently transfected with increasing amount of FLAG-praja2. Cells were harvested 24 h after transfection and lysed. Lysates were immunoblotted with anti-FLAG and anti-EGFR antibodies. **h** Quantitative analysis of experiment shown in (**g**). A mean value of three independent experiments ± SD is shown; Student's *t*-test for 2Υ*P = 0.02; Student's *t*-test for 4Υ *P = 0.04. **i** CaKi-1 cells were transiently transfected with increasing amount of FLAG-praja2. Cells were harvested 24 h after transfection and lysed. Lysates were immunoblotted with anti-FLAG and anti-EGFR antibodies. **j** Quantitative analysis of experiment shown in (**i**). A mean value of three independent experiments ± SD is shown.

compared to the normal counterparts (median 73.25 vs 80.05, student's *t*-test P = 1.6e-2) (Fig. 4f). Interestingly, we found a significant correlation between praja2 mRNA levels and survival rate of KIRC patients (long rank test, P = <1e4) (Fig. 4g). To corroborate this finding, we used the R2 Genomics analysis and visualization platform and evaluated the mRNA levels of praja2 in kidney tumors (EXPO project, n = 261) and normal counterpart tissues (Tsunoda's project, n = 24). The results revealed comparable praja2 mRNA levels in both experimental groups (median = 9.29 vs 9.16, one-way ANOVA test P = 0.08) (Supplementary Fig. 4).

**Praja2 expression is downregulated by oncomiRs**
The data above indicate that praja2 mRNA levels, in contrast to the protein, were stable in ccRCC samples, pointing to post-transcriptional mechanism(s) underlying praja2 downregulation in the tumor tissues. Recent evidence implicates a causal role of different microRNAs (miRNAs) in the

pathogenesis of several kidney diseases, including kidney cancer[53,54]. To dissect the mechanism of downregulation of praja2 in RCC, we measured the levels of praja2 in kidney cells transfected with different miRNAs previously identified to be overexpressed in ccRCC samples[55]. Among the miRNAs tested, we found that hsa-miR-155 and hsa-miR-210 displayed the most inhibitory effect on praja2 expression (Fig. 5a, b). The RNA-RNA interaction prediction algorithms (IntaRNA, http://rna.informatik.uni-freiburg.de/) identified a region within the 3′UTR of praja2 mRNA with a significant score for binding to hsa-miR-155, whereas no binding sites were found for hsa-miR-210 (Supplementary Fig. 5a). To confirm a direct interaction between hsa-miR-155 and the 3′UTR of praja2 mRNA, a luciferase reporter assay was performed. To this end, the entire 3′UTR mRNA sequence of praja2 was cloned in a vector downstream the open reading frame of the reporter gene firefly luciferase. A construct with an anti-sense-oriented 3′UTR mRNA of praja2 was used as negative control (Fig. 5c).

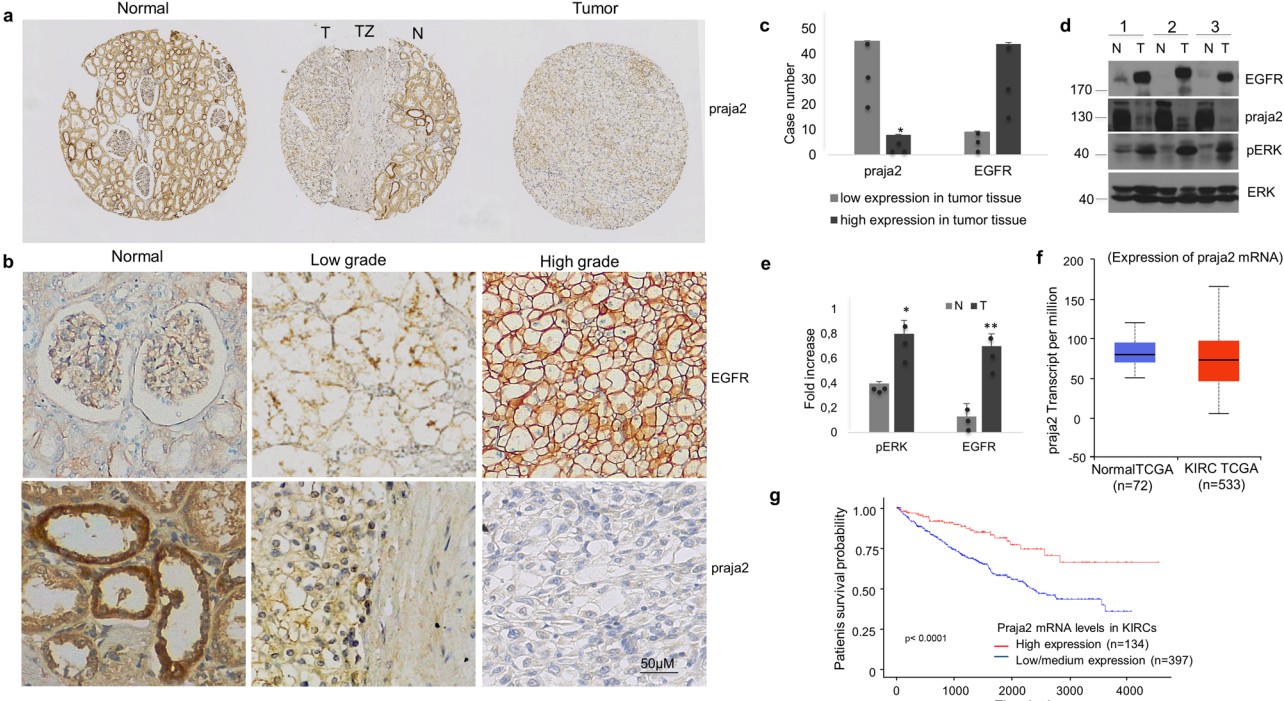

**Fig. 4 | Loss of praja2 in high-grade kidney cancer. a** Immunohistochemistry analysis of praja2 in normal renal tissue and renal tumor lesions. The middle section shows immunostaining for praja2 at the transition zone (TZ) between normal (N) and tumoral (T) areas. Magnification 40×. **b** Immunohistochemistry analysis for praja2 and EGFR. Normal tissue, low-grade tumor, and high-grade tumor are shown. EGFR expression is low in normal renal tissue and increased in clear cell renal cell carcinomas in a variable manner (upper panels). Inversely, praja2 expression is high in normal tissue and strongly decreased in tumors (lower panels). **c** Quantitative analysis of experiment shown in (**b**). **d** Immunoblot analysis of praja2,

EGFR, and phosphorylated ERK1/2 in total lysates prepared from ccRCC human tissues. **e** Quantitative analysis of experiment shown in (**d**). A mean value of three experiments ± SD is presented. Student's t-test for EGFR, *$P = 0.002$; Student's t-test for pERK1/2 *$P = 0.014$ (**f**) Boxplot showing relative expression (RNA transcript per million) of praja2 in normal samples and in primary lesions of kidney renal clear carcinoma (KIRC). UALCAN Database and Statistical Analysis Student's t-test were used ($P \leq 0.01$). **g** Kaplan–Meier plots showing the association between praja2 mRNA levels and patient survival (log-rank test, $P \leq 1e-4$).

Vectors (sense and anti-sense) and a control plasmid were transiently co-transfected with hsa-miR-155 in HEK293 cells and the luciferase activity was scored. Figure 5d shows that transfection of hsa-miR-155 markedly downregulated the luciferase activity in the sense-oriented 3′UTR praja2 mRNA vector, but not in the anti-sense-oriented construct, supporting a role of hsa-miR-155 in the silencing of praja2 expression. Downregulation of praja2 by hsa-miR-155 markedly impaired EGFR internalization both in EGF-treated HEK293 (Supplementary Fig. 5b) and HeLa (Fig. 5e–g), thus replicating the effects of praja2 silencing on EGFR turnover shown in Fig. 2. To confirm that praja2 downregulation by hsa-miR-155 was, indeed, the mechanism of impaired EGFR endocytosis, we performed rescue experiments in hsa-miR-155-transfected cells with praja2 transgene (either wild type or RING mutant). As shown in Fig. 5f, g expression of the wild-type praja2, but not its inactive mutant, markedly downregulated EGFR staining, compared to surrounding miR-155-transfected cells, that exhibit the EGFR staining prevalently localized on the cell membranes.

**Praja2 inhibits the growth and invasiveness of RCC cells**

The data above indicate that praja2 controls endocytosis and turnover of membrane receptors, including EGFR and TfR. As a consequence, praja2 stabilization might significantly reduce the concentrations of membrane receptors, thus affecting the growth and the fate of several cell types due to the variety of receptors involved. To clarify this issue, we assessed the biological role of praja2 in RCC cells by generating a kidney cancer cell line expressing FLAG-praja2 transgene (either wild type or inactive mutant) under the control of a doxycycline(dox)-inducible promoter. Treatment with the doxycycline induced, as expected, a time-dependent increase of praja2 levels which paralleled a significant decline of EGFR, whereas the levels of the same receptor were mostly unaffected by expressing the

catalytically inactive mutant praja2RM (Fig. 6a, b). Next, we monitored the biological effects of praja2 overexpression on cell growth. Figure 6c shows that kidney cancer cells expressing praja2 are mostly arrested at G0/G1 phase of the cell cycle. Furthermore, two independent clones of praja2 overexpressing A-498 present a marked reduction of the proliferation rate, compared to cells expressing praja2RM (Fig. 6d).

Amplification of EGFR signaling supports the motility and metastatic behavior of most epithelial cancer cells[56–58]. Given the role of praja2 in promoting downregulation of EGFR, overexpression of praja2 is expected to affect the metastatic potential of kidney cancer cells. We tested this hypothesis by performing zebrafish xenograft assay in embryos using the Tg (fli1:EGFP) zebrafish mutant line characterized by the expression of enhanced green fluorescence protein in the blood vessels throughout embryogenesis. The injection in perivitelline sack of fluorescently labeled cells and evaluation of their migration in other animal body parts was used to analyze in vivo the growth rate, the motility and the invasiveness behavior of a wide variety of cancer cells[59] (Fig. 6e). Accordingly, A-498 cells were fluorescently labeled with CM-DiL (red) tracker and then engrafted in 2 days-old zebrafish embryos Tg(fli1:EGFP). After the injection, the engraftment of injected tumor cells in the yolk sac was confirmed by fluorescence microscopy (Fig. 6f). We measured the motility of cancer cells from the site of injection to the head and tail of zebrafish embryos, both at 24 h and 72 h timepoints post-injection. As shown in Fig. 6f, g, the majority of injected embryos showed a high number of metastatic cells in the head and tail regions. In contrast, A-498 cells expressing praja2 (F11 clone) showed a significant reduction of invasive cells at both time points post-injection, compared to control and A-498 cells expressing praja2RM (E5 clone) (Fig. 6f, g). These findings indicate that praja2 inhibits the growth rate and motility of A-498 cancer cells in vivo.

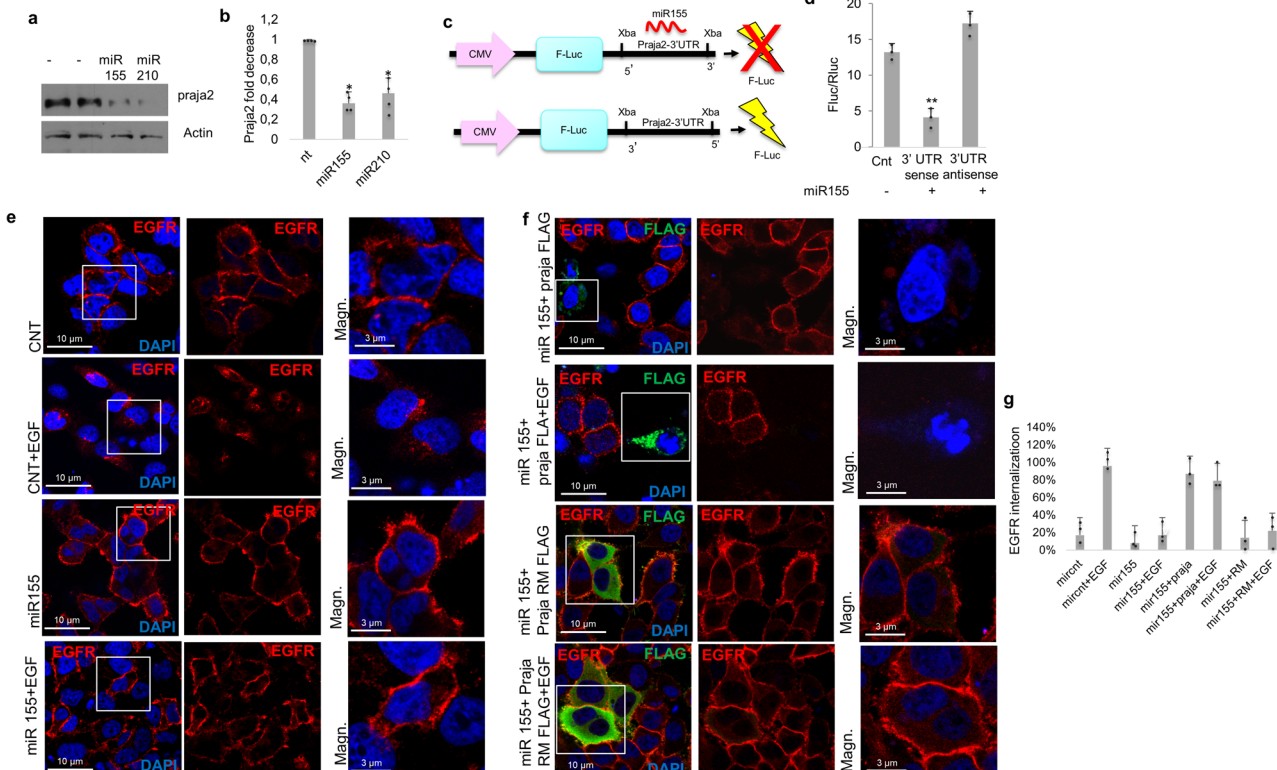

**Fig. 5 | Downregulation of praja2 by oncogenic miRNAs. a** Immunoblot analysis of HEK293 cells, transiently transfected with hsa-miR-155 and has-miR-210. Actin was used as loading control. **b** Quantitative analysis of the experiments shown in (**a**). A mean value of three independent experiments ± SD is presented. Student's *t*-test for hsa-miR-155 *P = 0.01; hsa-miR-210 *P = 0.013. **c** Schematic representation of the experiment shown in (**d**). **d** Relative luciferase activity and HEK293 cells transiently transfected with has-miR-155 oligonucleotides and, as control, a non-targeting scrambled oligonucleotide. The relative activity of firefly luciferase expression was standardized to a transfection control using Renilla luciferase. A mean value of three independent experiments ± SD is presented. Student's *t*-test

**0.0018. **e** HEK293 cells were transfected with, hsa-miR-155 or a scrambled oligonucleotide, serum-deprived overnight and treated with EGF (10 ng/µl) for 90 min. Cells were PFA-fixed and immunostained with anti-EGFR antibody (red) and DAPI (blue). Representative confocal images are shown. Scale bar is indicated. **f** Same as in (**e**), with the exception that a vector encoding for FLAG-praja2 (either wild type or RING mutant) was included in the transfection mixture. Representative confocal images are shown. Scale bar is indicated. **g** Quantitative analysis of the experiments shown in (**f**). The data are expressed as a mean value ± SD of internalized EGFR. About 50 cells for each experimental group of four independent experiments were analyzed.

## Transcriptional reprogramming in kidney cancer cells by praja2

Our data indicate that praja2 regulates the effective concentration of several receptors on the membrane and as consequence the signaling of growth factors, as observed in kidney cancer cells. To evaluate the actual impact of praja2 on gene networks, we performed RNA expression profiling of kidney cancer cells with inducible expression of praja2 by RNA sequencing (RNA-seq) analysis. For each RNA-seq experiment, two doxycycline-inducible clones of A-498 cells (D5 and F11) overexpressing FLAG-praja2 were used. On average, considering a normalized reads-count ≥10, 13300 transcripts were identified expressed in the biological replicates of the two clones. Comparing ± dox conditions, the expression of praja2 was 10.9-fold higher (*P*-adj = 1.88E-43) for clone D5 and 12.905 (*P*-adj = 6.69E-50) for clone F11. Differential expression analysis, normalized to ± dox conditions, shows 1022 downregulated (Fold-Change, FC ≤ −1.5; *P*-adj = ≤0.05) and 710 upregulated (FC ≥ 1.5; *P*-adj = ≤0.05) transcripts in the D5 clone and 974 transcripts downregulated (FC ≤ −1.5; *P*-adj = ≤0.05) and 888 upregulated (FC ≥ 1.5; *P*-adj = ≤0.05) in the F11 clone. Comparing the two datasets, we identified 458 downregulated (FC ≤ −1.5 in at least one clone; *P*-adj = ≤0.05) and 175 upregulated transcripts (FC ≥ 1.5 in at least one clone; *P*-adj = ≤0.05). Functional analysis performed on the dysregulated transcripts (633) revealed essentially two key cellular signal transduction pathways, oxidative phosphorylation and HER-2-signaling. Both pathways were significantly inhibited by praja2 expression (Z-score<2, *P* = < 0.05, Fig. 7a histogram and Fig. 7b, c Heatmaps). To detect representative genes downstream of EGFR regulation, RNA expression profiling was performed

using kidney tissues from wild-type (WT) and praja2 KO mice. The experiment was conducted considering three biological replicates per condition. Only one replicate for WT condition was excluded for further analysis due to a low alignment percentage on the reference genome. On average, considering a normalized reads-count cutoff ≥10, 16799 transcripts were identified as expressed. Analysis of differential expression indicates 543 downregulated (Fold-Change, FC ≤ −1.5; P-adj ≤ 0.05) and 904 upregulated (FC ≥ 1.5; P-adj ≤ 0.05) transcripts, that were used to perform pathways analysis. This functional annotation revealed the involvement of these transcripts in significant biological pathways, characterized by an activation Z-score > 2 or < −2 and a *P* ≤ 0.05 (Fig. 7d, e). The biological significance of the downregulation of genes involved in oxidative phosphorylation was confirmed by the metabolic profiling of A-498 cells. Thus, the oxygen consumption rate (OCR) of A-498 cells showed a predominant oxidative profile (Fig. 7f). Overexpression of praja2 in D5 and F11 A-498 clones markedly reduced the basal oxygen consumption rate and OCR-linked ATP production (Fig. 7f).

### praja2 knockout upregulates RTKs and induces epithelial and vascular proliferation in the kidney

To validate in vivo the data obtained in kidney cancer cells, we generated a praja2 knockout (KO) mouse model, by Cre/LoxP approach, targeting the exon2/3 region of the genomic locus with *a LoxP (L83)* site and a FNFL *(Frt-Neo-Frt-LoxP)* cassette (Fig. 8a). Genetically modified ES clones were isolated and injected into C57BL/6 J blastocysts. Genetically modified mice

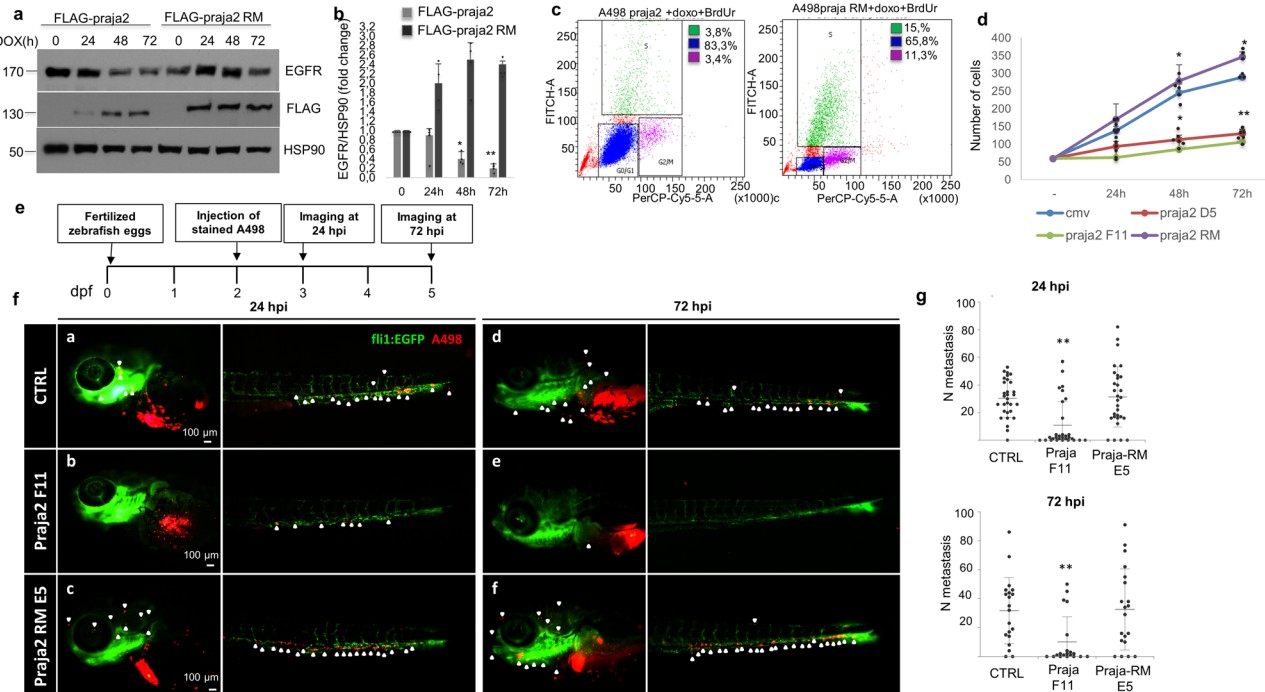

**Fig. 6 | Praja2 inhibits the growth and invasiveness of RCC cells. a** A-498 cells stably transfected with doxycycline-inducible vectors encoding for FLAG-praja2 or FLAG-praja2RM, were treated with doxycycline and harvested at the indicated timepoints. Lysates were immunoblotted with anti-FLAG and anti-EGFR antibodies. **b** Quantitative analysis of the experiments shown in (**a**). A mean value of three independent experiments ± SD is presented. Student's *t*-test **P = 0.0013; *P = 0.014. **c** FACS analysis of A-498 cells stably transfected with FLAG-praja2 or FLAG-praja2RM were treated with doxycycline for 48 h. Cell cycle distribution (G0/G1, S, and G2/M) is indicated as percentage of total cells scored. **d** A-498 cells stably transfected with FLAG-praja2 (two independent clones, D5 and F11), FLAG-praja2RM or empty vector (CMV) were seeded in multi-well plates and treated with doxycycline. Following the treatment, cells were collected and counted at different timepoints. A mean value of three independent experiments is shown. Student's *t*-test *P = 0.017 for FLAG-praja2 48 h; **P = 0.6e-3 for FLAG-praja2 72 h; *P = 0.01 for FLAG-praja2RM 48 h and 72 h; **e** Schematic view of the zebrafish model used in this study. **f** A-498-CMV, A-498-praja2, and A-498-praja2RM cells were injected into the perivitelline space (PVS) at 48 h post-fertilization (hpf) Tg(fli1:EGFP) zebrafish larvae. Cells were fluorescently labeled with CM-DiI (red) tracker and zebrafish tumor xenograft were analyzed at 24 h and 72 h post-injection (hpi) (a′, b′ and d′, e′ respectively). The metastatic cancer cells are indicated in zebrafish head and tail with white arrows. A representative image is shown. **g** Metastases in each zebrafish were counted 24 and 72 hpi. Distribution of animal with metastasis was presented as a dot blot. Student's *t*-test for 24 h, **P = 0.003; Student's *t*-test for 72 h, **P = 0.013.

were crossed with mice expressing the FLPe recombinase to remove the Neo cassette and then with *Hprt*/*Cre* mice to obtain the constitutive deletion of the *praja2* gene (Fig. 8a, b). We carried out preliminary studies on praja2+/− and praja2−/− mice by monitoring common phenotypic parameters such as body weight, growth, litter, appearance, behavior and fertility. No major differences were detected between wild-type and praja2 *KO* mice, except for a smaller body weight of praja2−/− mice compared to control and praja2+/− animals at Post-Natal Development 30 (PND30) (Supplementary Fig. 6). The difference in body weight eventually disappeared at later time point (PND60). At PND270, mice were sacrificed and kidneys were collected for further analysis.

Histological examination of fixed and paraffin-embedded sections of wild-type mice showed relevant histopathological changes in the kidney morphology (Fig. 8c). In praja2 *KO* mice, we observed the presence of dilated ectatic vessels that stood out as prominent, multifocal, and variable in size as open areas altering the normal renal architecture, both in the cortical and medullary regions. These dilatations contained a very slender and flattened endothelium (Fig. 8c), and they were associated with interstitial edema expanding the connective tissue framework evenly separating the tubules. The edematous fibrovascular stroma was focally and moderately infiltrated by chronic inflammatory cells such as lymphocytes and plasma cells. Renal glomeruli appeared diffusely and moderately atrophic with a slight multifocal and PAS-positive material in the mesangial matrix, widened Bowman's space, and shrinkage and hypocellularity of the capillary tufts (Supplementary Fig. 7). In one case, focal and severe cortical atrophy and slight mesangial hyperplasia were associated with moderate caliceal dilatation (hydronephrosis). Immunohistochemical analysis showed a

nuclear positivity both in the mesangial capillary tufts and endothelial cells of dilated vessels confirming that parenchymal dilatations were consistent with vascular enlargement and ectasia (Fig. 8c). In praja2 *KO* mice, we noticed a renal tubular ectasia with rarefaction of the nuclei that in some cases appear pyknotic, compatible with necroptosis. The presence of apoptotic and necrotic tissue explains the infiltration of immune cells. As expected, a significant upregulation of VEGFR and EGFR (Fig. 8d) and of the downstream oncogenic transcription factor ETS-related gene (ERG) (Fig. 8d) was evident in kidney section from praja2 *KO* mice, compared to control littermate, supporting the model of a negative regulation of RTKs signaling by praja2 in vivo.

## Discussion

Here, we report the identification of the RING ubiquitin ligase praja2 as a novel component of the endocytic pathway that negatively regulates the RTK signaling. Upon growth factors stimulation, praja2 interacts with- and recruits the adapter protein AP2 to the plasma membrane thereby promoting receptor-mediated endocytosis. Following growth factor stimulation, praja2 proteasomal-mediated degradation sustains RTKs stability and downstream signaling, thus supporting kidney cancer cells growth and invasion. Accordingly, genetic ablation of praja2 in mice markedly upregulates EGFR and VEGFR levels at the cell membrane, confirming the role of this ligase in RTKs turnover.

Renal cell carcinoma is the most common type of kidney cancer in the adult population. Therapy for RCC is still a burden for oncologists, since late diagnosis, resistance to therapeutics and disease recurrence account for the high mortality rate among RCC patients. Deregulated activation of tyrosine

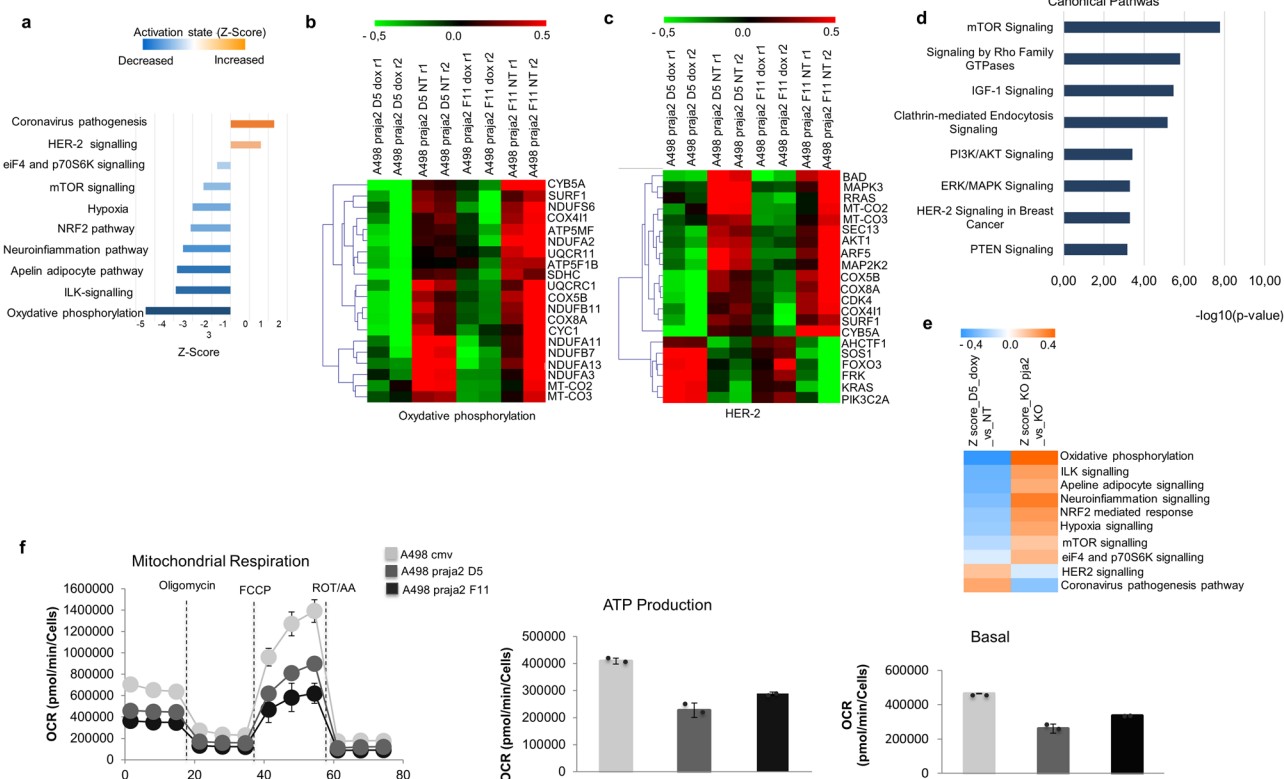

**Fig. 7 | Praja2 induces a transcriptional reprogramming in RCC cells.**
**a** Histogram showing activation state (Z-score values) of the canonical pathway statistically significant ($P = < 0.05$) involving the differentially expressed RNA transcripts, as computed by the IPA tool. **b, c** Heatmaps summarizing the expression values of the differentially expressed transcripts involved in the molecular signature of the indicated pathways, as computed with IPA, in two different clones of A-498 cells stably expressing praja2 (D5 and F11). The analysis was conducted in cells treated with doxycycline versus untreated cells. Data are shown as normalized expression values in log2 scale and centered on the median value. **d** Functional enrichment analysis enriched by IPA of the differentially expressed RNA transcripts identified in the dataset KO praja2 vs WT. Only pathways with $P ≤ 0.05$ were considered for the further analysis. **e** Heatmap showing the activation state, measured as Z-score values, of the canonical pathways statistically significant ($P ≤ 0.05$), involving differentially expressed RNA transcripts in Doxy vs NT and KO praja2 vs WT sets, respectively. **f** Real-time oxygen consumption rate (OCR) of the two different

clones of A-498 cells stably expressing FLAG-praja2 (D5 and F11) compared to cells expressing empty vector was measured at 37 °C using a Seahorse XF Analyzer (Seahorse Bioscience, North Billerica, MA, USA). Cells were plated into specific cell culture microplates (Agilent, USA) at the concentration of $3 × 10^4$ cells/well, and cultured for the last 12 h in DMEM, 10% FBS, in the presence of doxycycline. OCR was measured in XF media (non-buffered DMEM medium, containing 10 mM glucose, 2 mM L-glutamine, and 1 mM sodium pyruvate) under basal condition and after the sequential addition of 1.5 μM oligomycin, 2 μM FCCP, and rotenone + antimycin (0.5 μM all) (all from Agilent). Indices of mitochondrial respiratory function were calculated from OCR profile: basal OCR (before addition of oligo-mycin), maximal respiration (calculated as the difference of FCCP rate and anti-mycin + rotenone rate) and ATP production (calculated as the difference between basal OCR and oligomycin-induced OCR). Reported data are the mean values ± S.E.M. of four measurements deriving from two independent experiments.

kinase receptors in epithelial tumors, including RCC, represents an important mechanism that cancer cells adopt to promote the growth and invasiveness potential of malignant lesions. In particular, increased RTKs levels and signaling, due to genetic amplification or increased receptor stability/activity, play a pathogenic role in a large variety of epithelial tumors[60]. In renal cell carcinoma, RTKs are often upregulated, but genetic rearrangements or mutations of the receptors occur only in a minority of cases[61–63]. Mechanistically, it has been shown that hypoxic pathways upregulate the transcription of RTKs, including EGFR and VEGFR. Moreover, inhibitors of the hypoxia-VEGFR axis are currently being used in clinical trials for RCC therapy[7,64].

Despite the scientific knowledge of RCC biology, the mechanism(s) underlying upregulation of RTKs in the majority of RCC lesions are, so far, poorly understood. Our findings contribute to fill this gap by defining a key step underlying the fate of activated RTKs in kidney cancer cells. We found that the praja2's scaffold and catalytic functions are both required for receptor endocytosis and ligand-induced downregulation of EGFR and VEGFR, thus contributing to downstream oncogenic signaling attenuation. The findings obtained in kidney cancer cells were replicated in mice carrying genetic deletion of praja2 gene. In particular, we found a marked

upregulation of EGFR, VEGFR, and ERG levels in kidney tissues from praja2 knockout mice. This immunophenotypic asset was coupled to significant morphological changes that recapitulate the pathologic features of kidneys carrying overactivated tyrosine kinase receptors[13]. At a mechanistic level, we found that during RTK stimulation praja2 dynamically interacts, recruits, and ubiquitinates the adapter protein AP2 at the cell membrane. Downregulation of praja2 abrogates the ubiquitination and targeting of AP2 to the cell membrane, preventing receptor endocytosis and inducing a marked accumulation of the receptors at the membrane (Fig. 8e). These data indicate a role for praja2 as E3 ligase and scaffold for AP2, controlling the ligand-induced recruitment of the adapter protein at cell membrane and facilitating the endocytic route of the activated receptors. Whether praja2 ubiquitination regulates the AP2 complex assembly, its stability, and/or its binding to other partners involved in the early steps of endocytic vesicles formation and/or endocytic trafficking needs further substantial analysis. The role of praja2 in endocytosis is more general since genetic silencing of praja2 markedly affected the transferrin-induced internalization of TfR, inducing accumulation of TfR at the cell membrane, even in the presence of the ligand. These observations suggest that praja2 could play a role in the iron homeostasis and iron-controlled metabolic pathways.

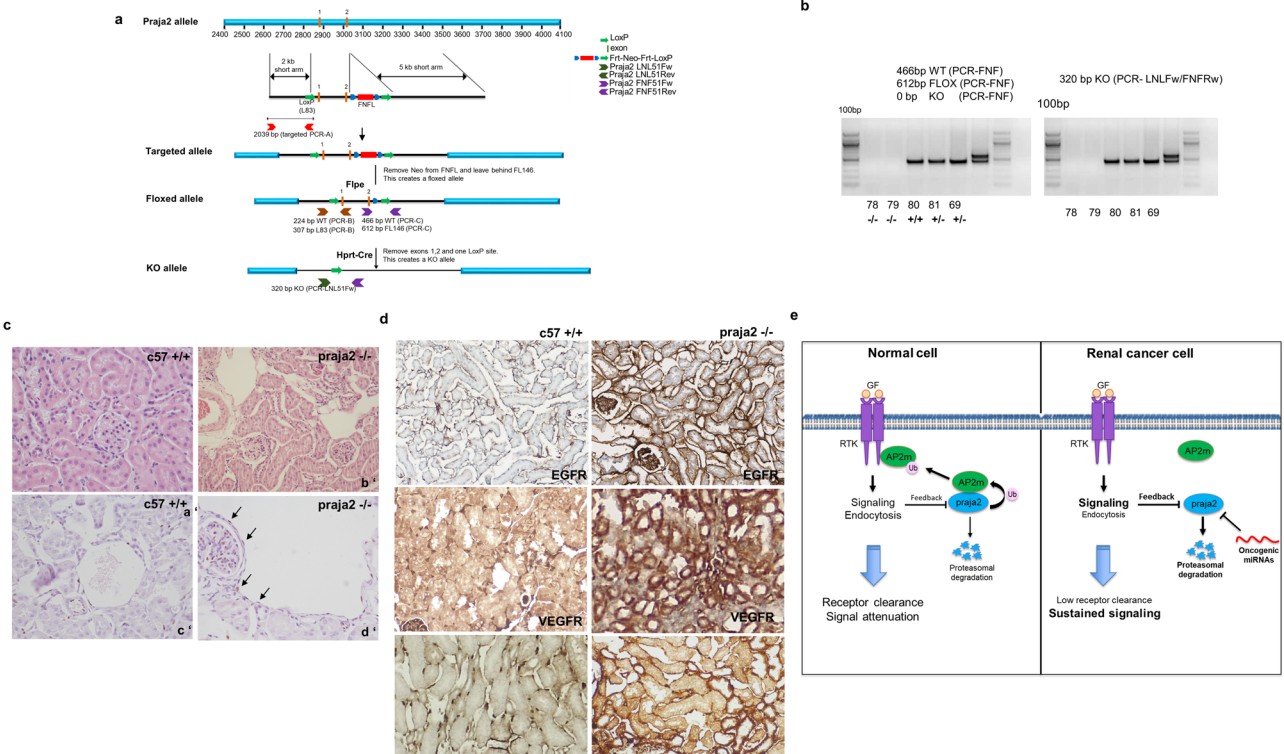

**Fig. 8 | Upregulation of RTKs in praja2 knockout (KO) mouse kidneys.**
**a** Schematic view of the targeted genomic locus of mouse praja2. Following deletion of the Neo Cassette by Flpe and Exon 1 and 2 by Hprt-Cre recombinases, respectively, the constitutive praja2 KO mouse line was obtained. **b** The offsprings from praja2[+/−] intercrosses were genotyped by PCR analysis of tail DNA. Bands corresponding to wild type (466 bp) and KO (320 bp) alleles were obtained using the couples of primers FNFFw/FNFRw and LNLFw/FNFRw, respectively. **c** Histological examination of kidney samples: hematoxylin/eosin (H/E) (a′, b′) and ERG-immunolabeled nuclei of endothelial cells (c′, d′) are shown. Magnification 40X. Wild-type kidney samples (a′–c′) show unremarkable histopathological changes. In praja2 KO kidneys (b′–d′), a prominent, ectatic dilatation of vessels bound only by a very slender and flattened endothelium was evident. Vascular enlargement and ectasia are bordered by ERG-immunolabeled nuclei (arrows) consisting of endothelium (d′). **d** Immunohistochemistry analysis for EGFR, VEGFR, and ERG in renal tissues from control (c57, left panels) and praja2 KO mice (79, right panels). Magnification 106×. **e** Growth factor binding to its cognate receptor (receptor tyrosine kinase, RTK) at cell membrane induces receptor activation and signaling. Ubiquitylation of the adapter protein AP2m1 by praja2 regulates receptor endocytosis and clearance, thus attenuating the mitogenic cascade. Downregulation of praja2 in clear cell carcinoma by oncogenic miRNAs and the proteasome impairs endocytosis and supports RTK signaling and cancer cell growth.

Our findings identified the existence of a negative feedback regulation of praja2 by RTKs that promotes praja2 proteolysis through the proteasome. Downregulation of praja2 in course of RTK activation may subserve, thus, as a mechanism to sustain downstream mitogenic and survival pathways in cancer cells, promoting growth and diffusion of malignant lesions. Several lines of evidence support this hypothesis. First, epithelial cancer cells expressing high levels of EGFR show undetectable amounts of praja2. Vice versa, high levels of praja2 in immortalized non-tumoral kidney cells (HEK293) are linked to low levels of EGFR. Second, regulated elevation of praja2 in kidney cancer cells induces a time-dependent decline of EGFR levels and inhibits the metastatic diffusion of cancer cells in vivo. Third, expression analysis in a wide variety of human kidney cancer tissues demonstrated an inverse correlation between the levels of praja2 and RTKs. In particular, high-grade renal cell carcinoma tissues express high levels of EGFR which paralleled an increased activation of downstream mitogenic kinases. In contrast, low levels of praja2 and phospho-ERK1/2 were observed in low-grade kidney cancer samples and in normal tissues. Our data also add another level of complexity in the regulation of praja2 levels in cancer cells by demonstrating that praja2 mRNA is a direct target of oncogenic miRNAs. miRNAs are non-coding single-stranded RNAs that complementary match mRNAs sequences of a variety of gene transcripts, markedly impacting on the expression of genes involved in many cellular pathological processes, including cancer growth and dissemination[65,66]. We identified a nucleotide binding sequence within the 3′-UTR of praja2 mRNA for the oncogenic miR-155,

a miRNA found to be upregulated in RCC tissues and also in other epithelial cancer tissues. When expressed in cells, miR-155 inhibits endogenous praja2 accumulation and prevents ligand-induced endocytosis of EGFR. The role of other miRNAs in the endocytic trafficking and RTK turnover as well as their impact on cancer cell growth and metastasis have been previously reported[67–69]. However, to our knowledge, our work provides the first demonstration of the existence of a signaling circuitry that mechanistically links oncogenic miRNAs to the ubiquitin pathway and RTK turnover.

In summary, our findings identified an interlink between miRNAs and the ubiquitin system that controls RTK endocytosis and clearance in renal cancer cells. In course of receptor activation, the E3 ligase praja2 supports receptor endocytosis and signal attenuation. Downregulation of praja2 by the proteasome and oncogenic miRNAs deregulates the endocytic pathway, favoring accumulation of RTKs at cell membrane and uncontrolled activation of downstream cancer cell signaling. Understanding the regulatory circuitry underlying praja2 downregulation in epithelial cancer cells and identifying the key elements involved will likely contribute to designing innovative therapeutic strategies for RCC treatment.

### Statistics and reproducibility
Each experiment was repeated three to five times as described in the legend figures. All results are expressed as mean ± SEM in dot plots. Data distribution and gene expression statistical analyses were performed using GraphPad Prism software (v5.0; GraphPad Software Inc., San Diego, CA),

Microsoft Excel 2016 (v16.04471; Microsoft Office 2016), and Interactive Dotplot (http://statistika.mfub.bg.ac.rs/interactive-dotplot/). Comparisons of two groups were performed using a Student's *t*-test. A *p*-value of <0.05 was considered to be statistically significant.

## Materials and methods

### Cell lines

Human embryonic kidney cell line (HEK293, ATCC), human cervical carcinoma cell line (HeLa, purchased from ATCC), and human kidney cell lines (A-498, HK-2, CaKi-1, purchased from ATCC) were cultured in Dulbecco modified Eagle's medium, containing 10% fetal bovine serum in an atmosphere of 5% $CO_2$. Human Kidney Carcinoma Cell Line (A-498) as well as A-498 overexpressing praja2WT or praja2RM under a tetracycline-inducible promoter, were maintained in DMEM medium (Gibco) supplemented with 10% Fetal Bovine Serum, NCS (Sigma-Aldrich), containing 1% L-glutamine and 1% penicillin (Sigma-Aldrich). Inducible lentiviral vectors, were generated using the In-Fusion® HD Cloning Kit technology. praja2 CDSs were amplified by PCR using Origene plasmids vectors as template and cloned in PCW57.1 vector (Addgene #41393) digested with Bam HI. A-498 cells were transfected by calcium-phosphate, using a DNA mix containing plasmid for over-expression (12 µg) harboring or not praja2 CDSs and psPAX2 (Addgene #12260) (18 µg) and pMD2.G (Addgene #12259) (13 µg) as packaging vectors. Forty-eight hours post-transfection, cell supernatants were collected, filtered through a 0.45 µM filter, and 10 µg/mL of polybrene (Merk) was added. A-498 cell media was then replaced with infective media, and cells were incubated for 6 h. Later, the infective cell medium was replaced with normal medium and, after additional 24 h, 1 µg/mL of puromycin was added to start selection. The day after the transfection, cells were induced or not, using 1 µg/ml of doxycycline, for 72 h, then cells were collected and lysed for Western blotting analysis.

### Plasmids, transfections, and Luciferase assays

Vector encoding for praja2 was purchased from Genecopeia (Rockville, USA), all the deletion mutants and the RM were generated as previously described[32]. AP2-HA vector was purchased from Genescript Biotech (New Jersey, USA). Myc tagged ubiquitin was previously described[32]. SMART pool siRNA directed against different segments of praja2 coding sequence was purchased from Dharmacon. The following are the siRNA sequences (Thermo Scientific; LU-006916-00- 10) targeting human praja2:

sequence 1: 5′-GAAGCACCCUAAACCUUGA-3′;
sequence 2: 5′-AGACUGCUCUGGCCCAUUU-3′;
sequence 3: 5′-GCAGGAGGGUAUCAGACAA-3′;
sequence 4: 5′-GUUAGAUUCUGUACCAUUA-3′.

The 3′-UTR region of *praja2* gene, including binding site for miR-155, was amplified from HEK293 cells by using the following primers:

3′-UTR-XbaI-Fw CTAGTCTAGAGAGATCAGTGTATCAAAGTA AAT;
3′-UTR-XbaI-Rev CTAGTCTAGAGACTCCTTGACACTAACTGA TC.

The amplified fragment was cloned into pGL3-Control firefly luciferase reporter vector (Promega) at the XbaI site. HEK293 cells were co-transfected with the pGCL3- Control firefly luciferase reporter vector with the praja2 3′UTR, the *Renilla* luciferase reporter plasmid and with the hsa-miR-155. Firefly and *Renilla* luciferase activities were measured 48 h after transfection with the Glomax luminometer microplate reader. Firefly activity was normalized to *Renilla* activity to control the transfection efficiency.

### Antibodies and chemicals

The following primary antibodies were used: hemagglutinin epitope HA.11 (catalog #16B12, IP dilution 1:200; immunoblot dilution 1:1000) from BioLegend; FLAG epitope (catalog #F3165, IP dilution 1:200; immunoblot dilution 1:2000) from Merck; praja2 (catalog A302-991A, IHC dilution 1:100, immunoblot dilution 1:1000) from Bethyl Laboratories; EGFR (catalog 06-847, IF dilution 1:500, immunoblot dilution 1:1000, IHC dilution 1:200) from Merck; TFR (catalog 13-6800, IF dilution 1:100; immunoblot dilution 1:1000) from Invitrogen; AP2m (catalog pa5_85381, immunoblot dilution 1:1000) from Invitrogen; HSP90 (catalog#13171-1, immunoblot 1:5000) from Proteintech; alpha-tubulin (catalog #T6199, immunoblot 1:5000) from Merck; VEGFR (catalog 9698, IHC dilution 1:100) Cell Signaling.

### Interactome analysis of praja2

HEK293 cells overexpressing FLAG-praja2RM were harvested and lysed in a dedicated buffer (1% Triton X-100, 150 mM NaCl, 1 mM EDTA, 50 mM Tris-HCl pH 7.5, containing protease inhibitors, PMSF and phosphatase inhibitors)[39]. Lysates were immunoprecipitated with anti-Flag M2 affinity gel (Merck, # A2220) for 3 h. After three washes with the above-reported buffer, proteins were eluted by incubation with 3xFlag-peptide (Thermo-Fisher, #A36805) (150 ng/µl) in PBS, for 2 h. Lysates from HEK293 cells overexpressing Flag-empty vector were immunoprecipitated in parallel and used as control. Both immunopurified protein samples were analyzed by 10% T SDS-PAGE and stained with colloidal Coomassie blue. Whole gel lanes were cut into 15 slices, minced, and washed with water; gel portions were treated to allow protein *in-gel* reduction, S-alkylation with iodoace-tamide, and digestion with trypsin, as previously reported[70]. Protein digests were analyzed with a nanoLC-ESI-Q-Orbitrap-MS/MS platform consisting of an UltiMate 3000 HPLC RSLC nano system linked to a Q-ExactivePlus mass spectrometer through a Nanoflex ion source; all devices were from Thermo Fisher Scientific. Peptides were resolved on an Acclaim Pep-MapTM RSLC C18 column (150 mm × 75 µm ID, 2 µm particles, 100 Å pore size) (Thermo Fisher Scientific), and eluted with a gradient of solvent B (19.92/80/0.08 v/v/v water/acetonitrile/formic acid) in solvent A (99.9/0.1 v/v water/formic acid), with a flow rate of 300 nl/min. The gradient of solvent B started at 3%, increased to 40% over 40 min, raised to 80% over 5 min, remained at 80% for 4 min, and finally returned to 3% in 1 min, with a column equilibrating step of 30 min before the subsequent chromato-graphic run. The mass spectrometer operated in data-dependent mode using a full scan (*m/z* range 375–1500, a nominal resolution of 70,000, an automatic gain control target of 3,000,000, and a maximum ion target of 50 ms), followed by MS/MS scans of the 10 most abundant ions. MS/MS spectra were acquired in a scan *m/z* range 200–2000, using a normalized collision energy of 32%, an automatic gain control target of 100,000, a maximum ion target of 100 ms, and a resolution of 17,500. A dynamic exclusion value of 30 s was also used. Triplicate analysis of each sample was performed to increase the number of identified peptides. MS and MS/MS raw data files per lane were merged for protein identification into Proteome Discoverer v. 2.4 software (Thermo Scientific), enabling the database search by Mascot algorithm v. 2.6.1 (Matrix Science, UK). The following para-meters were used: UniProtKB human protein database (11/2020, 214889 sequences), including the most common protein contaminants; carbamidomethylation of Cys as fixed modification; deamidation of Asn and Gln, oxidation of Met, and pyroglutamate formation of Gln as variable modifications. Peptide mass tolerance and fragment mass tolerance were set to ±10 ppm and ±0.05 Da, respectively. Proteolytic enzyme and maximum number of missed cleavages were set to trypsin and 2, respectively. Protein candidates assigned on the basis of at least two sequenced peptides and Mascot score ≥30 were considered confidently identified. Definitive peptide assignment was always associated with manual spectra visualization and verification. Results were filtered to 1% false discovery rate. A comparison with results from the corresponding control sample allowed to identify contaminant proteins that, nonetheless their abundance, were removed from the list of praja2-interacting protein partners (Supplementary Data 1). Interactomic data have been deposited to the ProteomeXchange Con-sortium via the PRIDE partner repository with the dataset identifier PXD[71]. The mass spectrometry proteomics data have been deposited to the Pro-teomeXchange Consortium via the PRIDE partner repository with the dataset identifier PXD034966.

## Protein-protein interaction (PPI) network

Identified praja2-interacting proteins were analyzed using the inBio Discover web tool (ref) (inBio Map Data Version 2021_04_07) to build a protein-protein interaction (PPI) network based on highly trusted interactions derived from experimental evidence as well as pathways and other curated resources. The following setting was applied: Network Expansion disabled. The praja2 PPI network was represented as an undirected graph (i.e., nodes and edges symbolize proteins and interactions between them, respectively). The network functional enrichment analysis was performed using the Selected Pathways Annotations & Enrichment method available in inBio Discover, selecting the following Gene Ontology terms: Endocytosis (GO:0006897), Endoplasmic reticulum to Golgi vesicle-mediated transport (GO:0006888), Golgi vesicle transport (GO:0048193), Regulation of autophagy (GO:0010506) and Vesicle-mediated transport (GO:0016192).

## Co-immunoprecipitation and western blot analysis

Cells were lysed in saline buffer 1% v/v Triton (NaCl, 150 mM; Tris-HCl, 50 mM, pH8; EDTA, 5 mM) or, for immunoprecipitation assay, in saline buffer 0.5% NP40 (50 mM Tris-HCl, pH 7.4, 0.15 M NaCl, 100 mM EDTA,0.5% NP40) containing aprotinin (5 µg/ml), leupeptin (10 µg/ml), pepstatin (2 µg/ml), 0.5 mm PMSF, 2 mM orthovanadate, and 10 mM NaF. Lysates were clarified by centrifugation at 18.000 rcf for 10 min and subjected to immunoprecipitation with the indicated antibody at the indicated dilution. The precipitate samples and aliquots of whole cell lysates (50 µg) were resolved on sodium dodecyl sulfate (SDS) polyacrylamide gel and transferred on nitrocellulose membrane for 7 min with a Transblot turbo system (BioRad, USA). Filters were blocked for at least 1 h at room temperature in Tween-20 TBS (TTBS) (TBS-Sigma, 0.1% Tween-20, pH 7.4) containing 5% nonfat dry milk. Reactive signals were revealed by ECL Star (EuroClone) or LiteAblot extended (EuroClone).

## Immunofluorescence and confocal analysis

For immunofluorescence analysis, HeLa cells were plated on glass coverslips. Cells were fixed with 3% paraformaldehyde (PFA), permeabilized with 0,1% v/v Triton, and immunostained with the indicated antibodies. Immunofluorescence was visualized using a Zeiss LSM 510 Meta argon/krypton laser scanning confocal microscope (Oberkochen, Germany). Quantification of the immunofluorescent images and correlation (Pearson's) coefficient were calculated by the ImageJ software (NIH, Bethesda, MD, USA).

## Immunohistochemistry

All tumors were retrieved from the files of the Department of Pathology, University of Naples Federico II. All specimens were examined and the tumors were histologically graded by an expert uropathologist (LI) according to the guidelines of the World Health Organization[3]. Formalin-fixed, paraffin-embedded tissues from the tumors were selected. Representative slides of the tumors were stained with hematoxylin and eosin. Immunohistochemistry for praja2 was performed automatically with a Nexes instrument (Ventana). Antibody detection was performed using a multilink streptavidin–biotin complex method, and antibodies were visualized by a diaminobenzidine chromogen method. Negative control samples were incubated with primary antibodies only. Two tissue microarrays (TMA), containing 1.00 mm cores, were generated with a semi-automatic instrument (GALILEO CK3500) from selected areas of Formalin-fixed, paraffin-embedded tissues. Tissues, consisting of 54 clear cell renal cell carcinomas, 2 chromophobe renal carcinomas, 3 papillary renal carcinomas, 1 medullary renal carcinoma, 1 acquired cystic disease-associated renal cell carcinoma, and 2 renal oncocytomas, were obtained from patients that underwent to total or partial nephrectomy. TMA design was realized using IseTMA software. Normal and tumor core were in duplicate, and a normal-tumor transition zone core was present for each sample. 4 µM thick tissue sections of the cores were stained with anti-praja2 antibody (dilution 1:200). A score of 0 or 1 was considered as negative, and a score of 2 or 3 was considered positive.

## Gene expression data for association with tumors kidney

TCGA normal samples ($n = 72$) and Kidney renal clear carcinoma (KIRC, $n = 533$) primary tumors were analyzed with the UALCAN Platform. Statistical significance was evaluated with one-way ANOVA test. Another set of normalized gene expression data from kidney tumors ($n = 261$, GSE2109) was analyzed and compared to normal tissues ($n = 24$, GSE18674) with "R2: Genomics Analysis and Visualization Platform". Student's $t$-test was used to test the significant differential gene expression among groups. Association of gene expression levels with overall survival and event-free survival was tested by subdividing the individual gene expression profiles by median split into 'high' or 'low' expression groups, and Kaplan–Meier survival curves were plotted for each group. To evaluate the significant difference between the two groups, long rank test was used.

## Zebrafish experiments

Cell culture and labeling. A-498 cells of different experimental groups were labeled with red cell tracker CM-DiL (Thermo Fisher Scientific, Massachusetts, USA) according to manufacturer's instructions. Before the injection into zebrafish larvae, the cells were trypsinized, washed, and resuspended in PBS to obtain a cell suspension for zebrafish xenotransplantation. Animal experiments were performed in accordance with the European Council Directive 2010/63/EU and approved by Biogem s.c.ar.l. internal Ethics Committee (OPBA). Tg(fli1:EGFP) zebrafish line, with green fluorescent vessels, was raised, maintained, and paired under standard conditions. Zebrafish eggs were obtained from natural spawning and incubated in E3 medium at 28 °C for 48 h. Two days post-fertilization (dpf) embryos were dechorionated and anesthetized with 0.04% of tricaine (Merck) before cell microinjection. Approximately 100–200 cells/embryo, were injected in the perivitelline space of each embryo using a pneumatic PicoPump PV830 injector (World Precision instruments) equipped with an injection borosilicate glass needle (Sutter Instruments). 24 h post-injection (hpi), larvae with correct engraftment in the yolk sac were selected under Leica M205 FA fluorescence stereo microscope (Leica) and kept in an incubator at 34 °C for 72 h. Zebrafish larvae were anesthetized (as previously described) and evaluated at 24 and 72 h post-injection by fluorescence stereo microscope. Different filters were selected for fluorescence imaging and captured with Leica DFC450 C camera. Images of embryos at different stages of each experimental group were analyzed with ImageJ software.

## RNA sequencing analysis

Libraries preparation was performed as described previously[72]. In particular, RNA purity and integrity were assessed with a Nanodrop 2000c spectrophotometer (ThermoFisher Scientific, Waltham, MA, USA) and a 4200 TapeStation instrument (Agilent Technologies, Santa Clara, CA, USA), respectively. For RNA purity, an A260/280 ratio of 2.0 and an A260/230 ratio of 2.0–2.2 were considered acceptable; for RNA integrity, an RNA Integrity Number (RIN) of 9.0–10.0 has been obtained for all samples, indicating the absence of degradation and high integrity of RNA samples. For a precise estimation of the RNA concentration, a Qubit 2.0 fluorometer assay (Thermo Fisher Scientific, Waltham, MA, USA) has been employed. For RNA sequencing, 1 µg of high-quality total RNA was used for library preparation with a TruSeq Stranded Total RNA Sample Prep Kit (Illumina, San Diego, CA, USA) and sequenced (paired-end, $2 \times 75$ cycles) on the NextSeq 500 platform (Illumina, San Diego, CA, USA). For each experimental condition, two biological replicates were considered. Data analysis was performed as described previously[73]. In detail, the raw sequence files generated (.fastq files) underwent quality control analysis using FASTQC (http://www.bioinformatics.babraham.ac.uk/projects/fastqc/) and adapter sequences were removed using Trimmomatic version 0.38[74]. Filtered reads were aligned using STAR v2.7.9a with standard parameters[75]. For DOXY vs NT dataset, filtered reads were aligned on the human genome (assembly hg38) considering genes present in GenCode Release 37 (GRCh38.p13), while for KO Paja2 vs WT dataset, the sequences were aligned on mouse genome (release mm39), considering genes available in GenCode Release M32 (GRCm39). Quantification of expressed genes was performed using

featureCounts[76] and differentially expressed genes were identified using DESeq2[77]. A given RNA was considered expressed when detected by at least ≥10 raw reads. Differential expression was reported as |Fold-Change| (FC) ≥ 1.5 along with the associated adjusted $p$-value ≤ 0.05 computed according to the Benjamini–Hochberg method. Functional analysis on differentially expressed genes was performed using Ingenuity Pathway Analysis (IPA, Qiagen). Only Canonical Pathways and Molecular Function with a $p$-value < 0.05 were considered for further analysis. The RNA-seq row data are publicly available in ArrayExpress repository under accession number: E-MTAB-11900.

## Generation of a conditional praja2 CKO mouse line

A LoxP (L83) site and a FNFL (Frt- Neo-Frt-LoxP) cassette were engineered to flank exon 2/3 (2 Kb) of the praja2 allele to generate "floxed/neo" praja2 on a Bacterial Artificial Chromosome (BAC). A gene targeting vector was constructed by retrieving the 2 kb short homology arm (5′ to L83), the floxed sequence containing exon 2/3, the FNL cassette, and the 5 kb long homology arm (end of FNFL to 3′) into a plasmid vector carrying a DTA (Diphtheria Toxin Alpha chain) negative selection marker. The FNFL cassette confers G418 resistance during gene targeting in KV1 (129B6 hybrid) ES cells and targeted ES cells were confirmed by PCR and southern blotting analysis. One of the ES-positive clones was injected into C57BL/6 J blastocysts to generate chimeric mice. Chimeric founder male mice were bred with C57BL/6 J females for germline transmission and the consequent generation of a mouse line carrying the genetic locus of praja2 under the control of loxP sites (the authorization number for this study is: 2014/0015838). After the deletion of Neo Cassette by Flpe recombinase, praja2 floxed mice were crossed with Hprt-Cre mice for the excision of exon 1 and 2. The resulting heterozygous praja2 KO mice (praja2$^{+/-}$) were backcrossed into the C57BL/6 J background for three generations (F3). Praja2$^{+/-}$ mice were then interbred to produce homozygous praja2$^{-/-}$ animals. Genomic DNA was extracted[78] by ear clip biopsies and KO mice were genotyped by PCR using the following primers: FNFFw, 5′-GATGTTCCAAAGGAGAA-CACCTCAG-3′; FNFRw, 5′-TATGCAGGAAAAACACACTC GGTTC-3′; LNLFw, 5′-GAAAGGGTTGGTTTGGAGTACAGGT-3′. The wild-type fragment (466 bp) was generated by the primers FNFFw and FNFRw; the KO fragment (320 bp) was generated by the primers LNLFw and FNFRw. Amplifcation conditions were: 95 °C for 5 min; 35 cycles: 95 °C for 30 s, 62 °C for 30 s, 72 °C for 1 min; 72 °C for 10 min. Animals were group housed (five per cage), at a constant temperature (22 ± 1 °C) on a 12 h light/dark cycle (lights on at 7 AM) with food and water ad libitum.

## Measurements of real-time oxygen consumption rates

Real-time oxygen consumption rate of different A-498 clones were measured using a Seahorse XF analyzer (Seahorse Bioscience, North Billerica, MA, USA). Cells were plated into specific cell culture microplates (Agilent, USA) at the concentration of $3 \times 10^4$ cells/well, and cultured for the last 12 h in DMEM, 10% FBS, in the presence of doxycycline. Oxygen consumption rate was measured in XF media (non-buffered DMEM medium, containing 10 mM glucose, 2 mM L-glutamine, and 1 mM sodium pyruvate) under basal condition and after the sequential addition of 1.5 μM oligomycin, 2 μM FCCP, and 0.5 μM rotenone plus 0.5 μM antimycin (all from Agilent).

## Reporting summary

Further information on research design is available in the Nature Portfolio Reporting Summary linked to this article.

## Data availability

The RNA-seq raw data are publicly available in ArrayExpress repository under accession number: E-MTAB-11900 (Doxy and NT datasets) and E-MTAB-13105 (KO Praja2 and WT datasets). The mass spectrometry proteomics data have been deposited to the ProteomeXchange Consortium via the PRIDE partner repository with the dataset identifier PXD034966. All the uncropped blots, gels, and raw data are available within the article and its supplementary data files (Supplementary Fig. 8, Supplementary Fig. 9, and Supplementary Data 3). The mass spectrometry proteomics data have been deposited to the ProteomeXchange Consortium via the PRIDE partner repository with the dataset identifier PXD034966. All other relevant data and materials are available from the corresponding author on reasonable request.

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

## Acknowledgements
This work was supported by Fondazione AIRC per la Ricerca sul Cancro (grants IG2018-22062 and IG2023-29124 to AF, IG-23068 to AW), the Italian Ministry of University and Research (National Center for Gene Therapy and Drugs based on RNA Technology, PNRR-CN3: E63C22000940007; PRIN2022: E53D23009690006 to AF and PRIN2022: D53D23010570006 to GG), the European Regional Development Fund (POR Campania FESR 2014/2020, grant RarePlatNet CUP: B63D18000380007 and GENOMA e Salute CUP: B41C17000080007 to AF and AW). Assunta S. was supported by the Estee Lauder companies Italia fellowship of Fondazione AIRC, DB was supported by a fellowship by Fondazione di Medicina Molecolare e Terapia Cellulare (Ancona). Special thanks to Drs: Sara Pignatiello and Loredana Stasio for their technical assistance, Chyuan-Sheng Lin (Transgenic Mouse Facility at Columbia University NY), and Max Gottesman for support in generating praja2 knockout mice.

## Author contributions
LR, FC, ES, DB, RDD, SA, LL, and RI performed the experiments and analyzed the data. CDA and Andrea S performed proteomic experiments and analyzed the data. GG, Francesca R, and Assunta S performed the RNA-seq gene network analysis. ED analyzed the PPI subnetwork. MF and LI performed immunohistochemistry experiments and analyzed the data. CR and CA performed zebrafish experiments and analyzed the data. MS provides human kidney tumor tissues. NR, Filomena R, and CA generated the praja2 KO model and analyzed the phenotype. DDB and OP analyzed the kidney sections from praja2 KO mice. CG performed the confocal microscopy analysis. AF, LI, Andrea S, AW, and CA conceived and supervised the experiments, analyzed the data, and revised the manuscript. AF wrote the manuscript.

## Competing interests
The authors declare no competing interests.

## Ethics approval and consent to participate
The present research was conducted ethically in accordance with the World Medical Association Declaration of Helsinki. Written informed consent was obtained from participants for publication of the details of their medical case and any accompanying images. We have complied with all relevant ethical regulations for animal use under the guidelines of Italian Ministry of Health and the study was approved by the Animal Care Authorities (Authorization number: 2014/0015838 and 924/2017-PR del 29/11/2017).
