## [Peer Review File · Communications Biology]

Reviewers' comments:

Reviewer #1 (Remarks to the Author):

The presented study are very interesting.

However there are some issues which should be explained and verified:

1. I do not understand why the authors used HeLa cell line when they describe kidney cancer. The results presented in Fig1 and 2 should be repeated on renal cell line, Caki-1, Caki-2, or HK-2, RPTEC. HeLa cell line should no longer be used due to its highly altered genotype
2. The inverse correlation between EGFR and Praja 2 should be presented on different renal cell lines: tumor Caki-1, Caki-2, and normal, mature, non embryonic like HEK293, cells: HK-2, RPTEC.
3. What is the fate of AP2 protein after downregulation of praja2 by hsa-mir-155 presented in Fig5? Why did you use HeLa cell line in this experiment except of renal cell? ~What cell did you use in Fig5e? The rescue experiments should be performed for renal tumor cells like A498, Caki-1, Caki-2

Reviewer #2 (Remarks to the Author):

Rinaldi et al. identify the ubiquitin ligase Praja2 as an important regulator of receptor endocytosis and its loss supports kidney cancer. By gain and loss-of-function studies, they show that upon ligand stimulation Praja2 binds to AP2 and regulates receptor stability and signaling in RCC tissues and cultured cells. EGF treatment leads to Praja2-mediated polyubiquitination of AP2 leading to its proteasomal degradation. Increased Praja2 levels adapts cancer cell metabolism resulting in the inhibition of cancer cell growth and dissemination using zebrafish model. Transgenic mice deficient of praja2 show upregulated RTK-MAPK activation. They conclude that Praja2 functions as a novel tumor suppressor impacting on membrane receptors signaling.

The manuscript is well written and the data presented support the conclusions of the research which in general deserves publication. However, I find that minor adjustments are needed:

- For immunofluorescence analysis, quantification of the immunofluorescent images should be done.
- Fig 1C problem with magnification and ROI
- Lysates were clarified by centrifugation at 13000 rpm: specify the g.
- Negative control samples were incubated with primary antibodies only, this is not a suitable negative control.
- Fig 2B is not convincing as tubulin presented is overexposed and very variable.
- Fig 3A FLAG-praja2 RM should be ideally expressed at the same level, HSP90 not equal.
- Fig 5F miR 155+praja FLA+EGF - not representative, nucleus
- Fig6A - quantification does not correspond
- Fig8 - missing scale bar to images

Reviewer #3 (Remarks to the Author):

This manuscript offers the important new findings on the ubiquitin ligase praja2 with AP2m forms the AP2 adaptor complex, contributing to receptor endocytosis and clearance. Loss of praja2 upregulates RTKs and induces epithelial and vascular proliferation in RCC cells and in mice. In addition, the authors make the first demonstration of the existence of a signaling circuitry that mechanistically links oncogenic miRNAs to the ubiquitin pathway and RTK turnover. However, there are several shortcomings, some additional queries to address and some suggestions for improvement of the manuscript:

1. The author first demonstrated that praja2 ubiquitinated AP2m, and it is necessary to further clarify the category of ubiquitinated modification and whether Praja2 affects AP2m protein degradation.

2. The author first demonstrated that praja2 is required for receptor endocytosis, and then proposed a regulatory loop links EGFR and praja2. The author did not confirm how EGFR regulates praja2 in Figure 3, whether at the transcriptional or protein level? No reasonable explanation was provided during the discussion.

3. The author should establish a xenograft tumor model in mice using renal cancer cell lines with knockout or overexpression of praja2 to observe the effects on tumor proliferation and metastasis.

4. The addition of Figure 5 by the author does not seem to have any value for the entire study. It is unclear whether the combination of mir-155 inhibitors can enhance the therapeutic effect of EGFR inhibitors, which may enhance the clinical significance of this study.

5. The addition of Figure 5 by the author does not seem to have any value for the entire study. It is unclear whether the combination of mir-155 inhibitors can enhance the therapeutic effect of EGFR inhibitors, which may enhance the clinical significance of this study. If so, the author can detect representative genes downstream of EGFR regulation after praja2 knockout and clarify their regulation of the RTK signaling pathway.

Minor points:

1. The method for IHC quantification is not described in the method.

2. The WB band quality of Praja2 in the WCL group in Figure 1e is poor and needs to be revalidated

3. The icon in Figure 2c shows an error in the gene, which should be "siPraja2".

4. Statistical analysis needs to be added to the quantitative analysis of Figure 4e.

5. Statistical analysis needs to be added to the quantitative analysis of Figure 6g.

6. Statistical analysis needs to be added to the quantitative analysis of Figure 7d.

7. There are no 7e and 7f in Figure 7.

Reviewer #1 (Remarks to the Author):

The presented study are very interesting.

R. *Many thanks to the Reviewer to finding our manuscript “very interesting”*

However, there are some issues which should be explained and verified:

1. I do not understand why the authors used HeLa cell line when they describe kidney cancer. The results presented in Fig1 and 2 should be repeated on renal cell line, Caki-1, Caki-2, or HK-2, RPTEC. HeLa cell line should no longer be used due to its highly altered genotype

R. *We appreciate the Reviewer’s concern on the use of HeLa cells for our studies. Given the general role of praja2 in RTK turnover, HeLa cells were chosen because of their good levels of expression of both EGFR and praja2. In fact, praja2 levels are very low in kidney cancer cells (as shown in the **Fig. 3e**). Nevertheless, as suggested by the Reviewer, we repeated the experiments using other renal cells (HK2) and confirmed the colocalization between AP2m and praja2. Fig. 1c has been substituted with the **new Fig. 1c**. Furthermore, we performed the same experiment shown in the Fig. 2e using HK2 cells and obtained similar results (please, see the **new Supplementary Fig. 3a**).*

2. The inverse correlation between EGFR and Praja 2 should be presented on different renal cell lines: tumor Caki-1, Caki-2, and normal, mature, non embryonic like HEK293, cells: HK-2, RPTEC.

R. *As shown in the **new Supplementary Fig. 3a**, we confirmed that genetic silencing of praja2 induces an increase of EGFR levels also in HK2. Furthermore, we overexpressed praja2 in Caki1 cells and confirmed the inverse correlation between praja2 and EGFR (**new Fig. 3i**).*

3. What is the fate of AP2 protein after downregulation of praja2 by hsa-mir-155 presented in Fig5? Why did you use HeLa cell line in this experiment except of renal cell? ~What cell did you use in Fig5e? The rescue experiments should be performed for renal tumor cells like A498, Caki-1, Caki-2.

R. *All the experiments with has-mir-155 were performed in HeLa cells, including the experiment in the Fig. 5e, because in the renal tumor cell lines this particular miRNA is already overexpressed while praja2 is severely downregulated. These particular conditions make extremely difficult to perform rescue experiments in renal cancer cell lines. The EGFR levels, in fact, are decreased in A498 by stably overexpressing praja2 (please, see **Fig 6a**).*

Reviewer #2 (Remarks to the Author):

Rinaldi et al. identify the ubiquitin ligase Praja2 as an important regulator of receptor endocytosis and its loss supports kidney cancer. By gain and loss-of-function studies, they show that upon ligand stimulation Praja2 binds to AP2 and regulates receptor stability and signaling in RCC tissues and cultured cells. EGF treatment leads to Praja2-mediated polyubiquitination of AP2 leading to its proteasomal degradation. Increased Praja2 levels adapts cancer cell metabolism resulting in the inhibition of cancer cell growth and dissemination using zebrafish model. Transgenic mice deficient of praja2 show upregulated RTK-MAPK activation. They conclude that Praja2 functions as a novel tumor suppressor impacting on membrane receptors signaling. The manuscript is well written and the data presented support the conclusions of the research which in general deserves publication.

R. Many thanks to the Reviewer in finding our manuscript “well written and the data presented support the conclusions of the research which in general deserves publication”. We have addressed the minor comments raised as indicated below:

However, I find that minor adjustments are needed:

1. For immunofluorescence analysis, quantification of the immunofluorescent images should be done.

*R. The quantitative analysis of Fig. 2a, 2e is shown in the **new Fig. 2b, 2f**, while the quantitative analysis of Fig. 5e, 5f is presented in the **new Fig. 5g**.*

2. Fig 1C problem with magnification and ROI

*R. Figure 1c has been replaced with the **new Fig. 1c**.*

3. Lysates were clarified by centrifugation at 13000 rpm: specify the g.

R. We have clarified this point in the Materials & Methods section (please, see p27, lines 11-12).

4. Negative control samples were incubated with primary antibodies only, this is not a suitable negative control.

R. Negative controls were incubated with primary antibody overnight and for 45 minutes with A/G plus Sepharose beads. Precipitates were washed three times with 1% Triton buffer and loaded on the SDS-polyacrylamide gel.

5. Fig 2B is not convincing as tubulin presented is overexposed and very variable.

*R. The tubulin panel has been replaced (**new Fig. 2c**).*

6. Fig 3A FLAG-praja2 RM should be ideally expressed at the same level, HSP90 not equal.

R. We understand the point raised by the Reviewer. However, we wish to note that the inactive RING mutant of praja2 is more stable and has a longer half-life, compared to wild type protein. This a common feature of mutant inactive variants of most RING E3 ligases. Since transfection experiments long last for 48h, the mutant praja2 accumulates at greater levels than wild type protein.

7. Fig 5F miR 155+praja FLA+EGF - not representative, nucleus.

*R. A better quality nuclear staining has been included. Also, a quantitative analysis has been added in the **new Fig. 5g** that represents a mean value of internalized EGFR from about 50 cells for each experimental group.*

8. Fig6A - quantification does not correspond

*R. The quantitative analysis in the **Fig. 6b** represents the mean value of four independent experiments.*

9. Fig8 - missing scale bar to images

R. *We have included the magnification index in the legend to Fig. 8.*

Reviewer #3 (Remarks to the Author):

This manuscript offers the important new findings on the ubiquitin ligase praja2 with AP2m forms the AP2 adaptor complex, contributing to receptor endocytosis and clearance. Loss of praja2 upregulates RTKs and induces epithelial and vascular proliferation in RCC cells and in mice. In addition, the authors make the first demonstration of the existence of a signaling circuitry that mechanistically links oncogenic miRNAs to the ubiquitin pathway and RTK turnover. However, there are several shortcomings, some additional queries to address and some suggestions for improvement of the manuscript:

R. *We wish to thank the Reviewer for the positive comments on our manuscript and for the insightful suggestions. We have addressed his/her comments as below:*

1. The author first demonstrated that praja2 ubiquitinated AP2m, and it is necessary to further clarify the category of ubiquitinated modification and whether Praja2 affects AP2m protein degradation.

R. *As suggested by the Reviewer, we repeated the ubiquitination experiments using antibody directed against ubiquitinated lys48 and lys63 and we found only a slight increase of the ubiquitylated AP2 at these sites (Please, see the **new Supplementary Fig. 2**). Nevertheless, we wish to point that ubiquitylation of AP2 by praja2 is not degradative. Accordingly, as shown in the **new Fig. 2c**, genetic knock down of praja2 did not affect AP2 levels, compared to controls.*

2. The author first demonstrated that praja2 is required for receptor endocytosis, and then proposed a regulatory loop links EGFR and praja2. The author did not confirm how EGFR regulates praja2 in Figure 3, whether at the transcriptional or protein level? No reasonable explanation was provided during the discussion.

R. *The regulation exerted by EGFR on praja2 levels is at post-transcriptional level, since the experiments shown in the **Fig. 3c** were performed in the presence of the protein synthesis inhibitor cycloheximide. We have added this information in the figure legend.*

3. The author should establish a xenograft tumor model in mice using renal cancer cell lines with knockout or overexpression of praja2 to observe the effects on tumor proliferation and metastasis.

R. *As suggested by the Reviewer, we performed two independent experiments in xenograft tumor models in mice using renal cancer cell lines. Overexpression experiments with lentiviral vectors transducing wild-type praja2 or empty vector show that praja2 did not affect tumor growth, compared to controls (please, see the **Figure for the Reviewer only** below). To understand the lack of effects of the overexpressed praja2 on tumor growth, we compared the expression levels of praja2 in transiently transduced cells (before the injection) and in the excised tumor samples. We found that praja2 protein was expressed in transduced cells (**panel a**), whereas it was nearly absent in all tumor samples analyzed (**panel e**). This finding strongly suggests that transiently overexpressed praja2 most likely undergoes to proteolysis through the proteasome in actively growing cancer cells, that is consistent with the model proposed in the manuscript. If necessary, we might include these data in the Supplementary section.*

Figure legend. Xenograft analysis of A498 cells infected with lentiviral particles encoding for praja2. a) Immunoblot analysis of A498 and A498 transiently over-expressing pja2 after the lentiviral infection. **b)** *In vivo* evaluation of engraftment and growth capability of A498 infected with lentiviral vector encoding for praja2 or with empty vector. The growth of infected cells *in vivo* was evaluated by xenograft assay in CD1-nude females (N=4/group). $3 \times 10^6/200\mu\text{l}$ cells with matrigel (1:1 ratio) of both cell lines were injected subcutaneously in the right flank of mice. Tumor growth has been evaluated weekly by caliper starting at day 7 after the injection and carried out weekly. **c)** Tumor volumes of excised lesions at four weeks from engraftment shown in the graph have been determined with the following formula: $(\text{mm}^3) = [\text{length (mm)} \times \text{width (mm)}]^2/2$ where the width and the length are the shortest and the longest diameters measured. **d)** Body weight analysis of mice, evaluated weekly. **e)** Hematoxylin/eosin (H/O) and immunohistochemistry (IHC) analysis with anti-Flag antibody of the excised tumor lesions.

4. The addition of Figure 5 by the author does not seem to have any value for the entire study. It is unclear whether the combination of mir-155 inhibitors can enhance the therapeutic effect of EGFR inhibitors, which may enhance the clinical significance of this study.

R. *We understand the comment raised by the Reviewer. We wish to point that the regulation of praja2 by mir-155, mir-210, and perhaps by other yet identified miRNAs, only contributes to a better understanding of the mechanisms of praja2 downregulation in kidney cancers to which concurs not only the proteasome but also the cancer-related miRNAs. Of course, further investigation is required to define the translational relevance of these findings.*

5. The addition of Figure 5 by the author does not seem to have any value for the entire study. It is unclear whether the combination of mir-155 inhibitors can enhance the therapeutic effect of EGFR inhibitors, which may enhance the clinical significance of this study. If so, the author can detect representative genes downstream of EGFR regulation after praja2 knockout and clarify their regulation of the RTK signaling pathway.

R. *Thanks to the Reviewer for this important suggestion. To address this comment, we performed a whole transcriptomic analysis using kidney tissues from wild type and Praja2 KO mice. The experiments were performed considering three biological replicates per condition. Only one replicate for the WT condition was excluded for further analysis due to a low alignment percentage on the reference genome. Considering a normalized reads-count cutoff ≥ 10 , 16799 transcripts were identified as expressed. Analysis of differential expression indicates 543 down-regulated ($\text{Fold-Change, FC} \leq -1.5$; $P\text{-adj} \leq 0.05$) and 904 up-regulated ($\text{FC} \geq 1.5$; $P\text{-adj} \leq 0.05$) transcripts). This data set was used to perform the functional analysis, that revealed their involvement in significant biological pathways characterized by an activation Z-score > 2 or < -2 and a $P \leq 0.05$. The analysis shown in the **new Figs. 7d, 7e** indicates the upregulation of EGFR target genes in KO tissues, compared to wild type. The results are coherent with the model proposed.*

Minor points:

1. The method for IHC quantification is not described in the method.

R. *We have included the IHC quantification*

2. The WB band quality of Praja2 in the WCL group in Figure 1e is poor and needs to be revalidated

R. *A better quality gel has been included.*

3. The icon in Figure 2c shows an error in the gene, which should be “siPraja2”.

R. *We have corrected the gene name.*

4. Statistical analysis needs to be added to the quantitative analysis of Figure 4e.

R. *Included.*

5. Statistical analysis needs to be added to the quantitative analysis of Figure 6g.

R. *Included.*

6. Statistical analysis needs to be added to the quantitative analysis of Figure 7d.

R. *Included.*

7. There are no 7e and 7f in Figure 7.

R. *Corrected.*

REVIEWERS' COMMENTS:

Reviewer #1 (Remarks to the Author):

I appreciate the authors for addressing my queries and responding to my comments.

Reviewer #2 (Remarks to the Author):

Authors have addressed the remarks and improved the manuscript.

Reviewer #3 (Remarks to the Author):

Comments to the Author
Manuscript COMMSBIO-23-0523A

Title: " Downregulation of praja2 restrains endocytosis and boosts tyrosine kinase receptors in kidney cancer"

Authors: Rinaldi et al.

The revised manuscript has been carefully revised and improved according to the previous review comments.